# U.S. West Coast droughts and heat waves exacerbate pollution inequality and can evade emission control policies

Amir Zeighami [1], Jordan Kern [1] ✉, Andrew J. Yates[2], Paige Weber[2] & August A. Bruno[2]

Droughts reduce hydropower production and heatwaves increase electricity demand, forcing power system operators to rely more on fossil fuel power plants. However, less is known about how droughts and heat waves impact the county level distribution of health damages from power plant emissions. Using California as a case study, we simulate emissions from power plants under a 500-year synthetic weather ensemble. We find that human health damages are highest in hot, dry years. Counties with a majority of people of color and counties with high pollution burden (which are somewhat overlapping) are disproportionately impacted by increased emissions from power plants during droughts and heat waves. Taxing power plant operations based on each plant's contribution to health damages significantly reduces average exposure. However, emissions taxes do not reduce air pollution damages on the worst polluting days, because supply scarcity (caused by severe heat waves) forces system operators to use every power plant available to avoid causing a blackout.

Despite decades of air quality improvement, air pollution in the United States is still associated with 100,000–200,000 premature deaths per year. Over 50% of this air pollution comes from the combustion of fossil fuels; of that amount, 10% is produced at electric power plants[1]. But air pollution from the power sector (and its cost to society) is not constant. Hydrometeorology, including extreme events like droughts and heatwaves, is known to be a significant driver of emmisions from power plants[2]. This is especially true in California's electric power system. Drought reduces the availability of water for in-state hydropower production, which meets 13.9% of California's annual electricity demand on average, with imported hydropower from the Pacific Northwest (PNW) providing another 2.5% of California's energy needs[3]. Heat waves increase electricity demands for cooling. Both phenomena force grid operators to rely more on fossil fuel power plants, resulting in increased emissions of greenhouse gases and other pollutants, including sulfur dioxide ($SO_2$), nitrogen oxides ($NO_x$) (precursors to the formation of ground

level ozone) and fine particulate matter ($PM_{2.5}$)[4–8], which have been shown to cause human health problems such as cardiopulmonary and lung cancer mortality[9].

In this paper, we show how droughts, heatwaves, and the spatial (county level) distribution of local air pollution emitted by power plants are linked via short-term dynamics in power systems, including shifts in the supply, demand, and flow of electricity on large geographic scales. Given the vast scale and interconnected nature of the West Coast grid, droughts and heat waves affecting one spatial domain (e.g., the PNW) can prompt increased emissions at power plants in another (e.g., California). Droughts and heat waves may also have important ramifications for who is impacted. Even under "normal" weather conditions, air pollution from power plants disproportionately affects marginalized socio-economic groups and people of color[10]. An open question is whether droughts and heat waves, by triggering additional pollution from power plants, exacerbate this inequity.

[1]Department of Foresry and Environmental Resources, North Carolina State University, Raleigh, NC, USA. [2]Department of Economics, University of North Carolina-Chapel Hill, Chapel Hill, NC, USA. ✉e-mail: jkern@ncsu.edu

Droughts and heatwaves may also evade policies aimed at curbing human health damages from power plants, such as financial penalties (i.e. taxes) levied on emissions. During normal operating conditions, these policies incentivize system operators to use lower polluting plants first by increasing the marginal cost of heavier polluters[11,12]. However, extreme droughts and heat waves may force system operators in California to turn on nearly all available power plants, including heavy emitters, in order to meet electricity demand and avoid rolling blackouts. During these events, financial penalties levied on emissions may not result in reduced health damages from power plants. If emissions penalties are less effective at reducing air pollution damages during droughts and heat waves, previous estimates of the costs and benefits of such pollution control policies could be inaccurate. At present, there has been no rigorous weather "stress" testing of emissions control policies for power plants.

This paper aims to explore the effects of droughts, heat waves, and hydrometeorological uncertainty more broadly on human exposure to power plant emissions, focusing on California as a case study. Using an open source, stochastic power system simulation tool (the CAPOW model) in conjunction with an integrated assessment model, we model power plant emissions of $SO_2$, $NO_X$, and $PM_{2.5}$ and the monetary cost of associated human health damages in California under 500 synthetic weather years and multiple emissions penalty scenarios. For the first time, we track how spatially explicit anomalies in hydrometeorological conditions across the West Coast translate to county-level health damages in California for different demographic groups. We also explore the potential for drought and heat waves to create days with extremely high levels of air pollution, which may be unaffected by policy interventions. Our results indicate that dry years and hot years exacerbate existing inequalities in pollution damages, with increases in damages being concentrated in counties with larger people of color population shares, and in counties that already bear a higher pollution burden. Although they can occur in tandem, droughts and heat waves pose very distinct air quality risks for California. Drought increases chronic exposure over several months due to lost hydropower generation, but heat waves (which cause shorter term spikes in high electricity demand) cause the most severe days. Although rare, we do find that during severe heat waves, when emissions from power plants in California are highest, financial penalties on

power plant emissions do not alter power plant operations or resulting air pollution damages, leaving people unprotected from acutely high damage days.

## Results

### Uncertainty characterization
Figure S1 in the Supplemental Information section illustrates our modeling approach and experimental setup. Our results are based on a 500-year stochastic ensemble of hydrometeorological data consisting of daily streamflow (85 observation stations), air temperatures (17 observation stations), wind speeds (17 observation stations), and solar irradiance values (7 observation stations) across the West Coast. Annual streamflow and air temperature values are jointly sampled using a Gaussian copula approach and historical data from 1953 to 2008, with daily streamflow "fractions" resampled from historical years based on spring air temperatures (a proxy for snowmelt timing). Daily air temperatures, wind speeds and solar irradiance are modeled stochastically as deviations from average 365-day profiles, with correlations in these deviations across variables and space captured using a vector autoregressive model fitted using historical data from 1998 to 2017. Streamflow, air temperature, wind speed, and solar irradiance data are then translated to values of hourly electricity demand, daily availability of hydropower, and hourly availabilities of wind power and solar power across the system. These time series are used to force a model of the 2018 version of the West Coast bulk electric power system (see Supplementary Fig. S2)[6,13], which produces zonal market prices and hourly estimates of electricity generation and emissions at each generator. Time series of emissions are then linked to health damages using the Air Pollution Emission Experiments and Policy integrated assessment model (AP3).

Figure 1 examines multi-scale uncertainties in selected state variables and performance metrics for the 500-year stochastic ensemble. Figure 1a, b shows daily distributions of electricity demand and hydropower availability, respectively, in the California Independent System Operator (CAISO) system, which manages approximately 80% of electricity flow in California. There are strong seasonal trends in median conditions, and electricity demand in particular exhibits seasonal trends in volatility due to the non-linear relationship between air temperatures and electricity demand. Hydropower production peaks

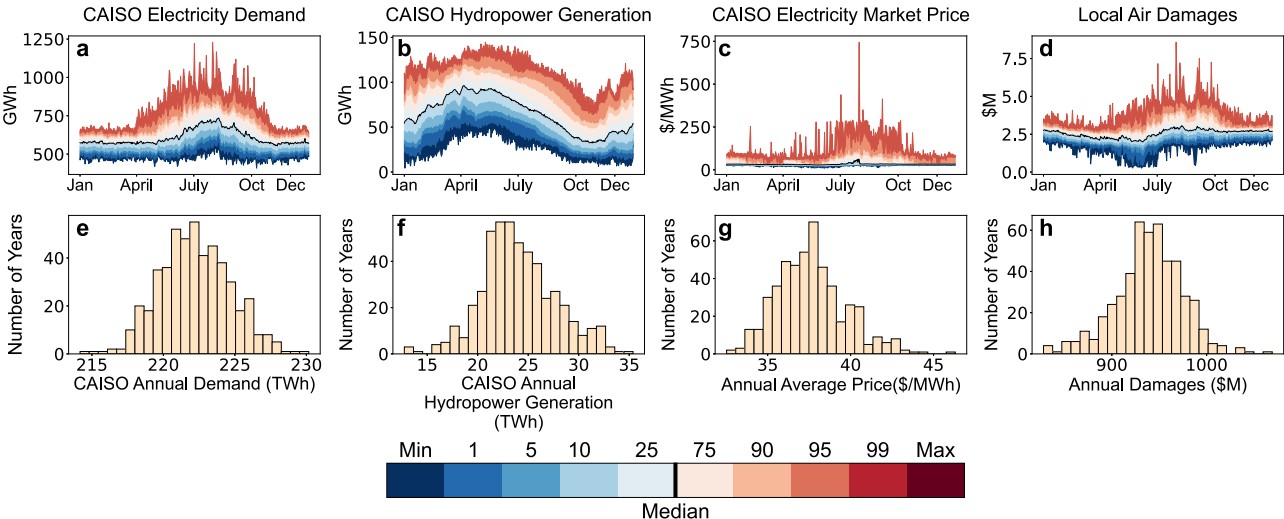

**Fig. 1 | Simulated annual and seasonal uncertainty in selected system variables (electricity demand and hydropower production) and performance metrics (CAISO electricity market prices and local air damages from $SO_2$, $NO_X$, and $PM_{2.5}$ in California).** The first row shows daily distributions produced using the full 500-year stochastic weather ensemble. Panels shown are (**a**) CAISO electricity demand, (**b**) CAISO hydropower production, (**c**) CAISO electricity market price; and (**d**) local air damages in California. Seasonal patterns in electricity demand and hydropower production are key drivers of dynamics in market prices and local air pollution damages. The second row shows distributions of annual average values for the same 500-year ensemble. Panels shown are: (**e**) CAISO electricity demand; (**f**) CAISO hydropower production; (**g**) CAISO electricity market price; and (**h**) local air damages in California.

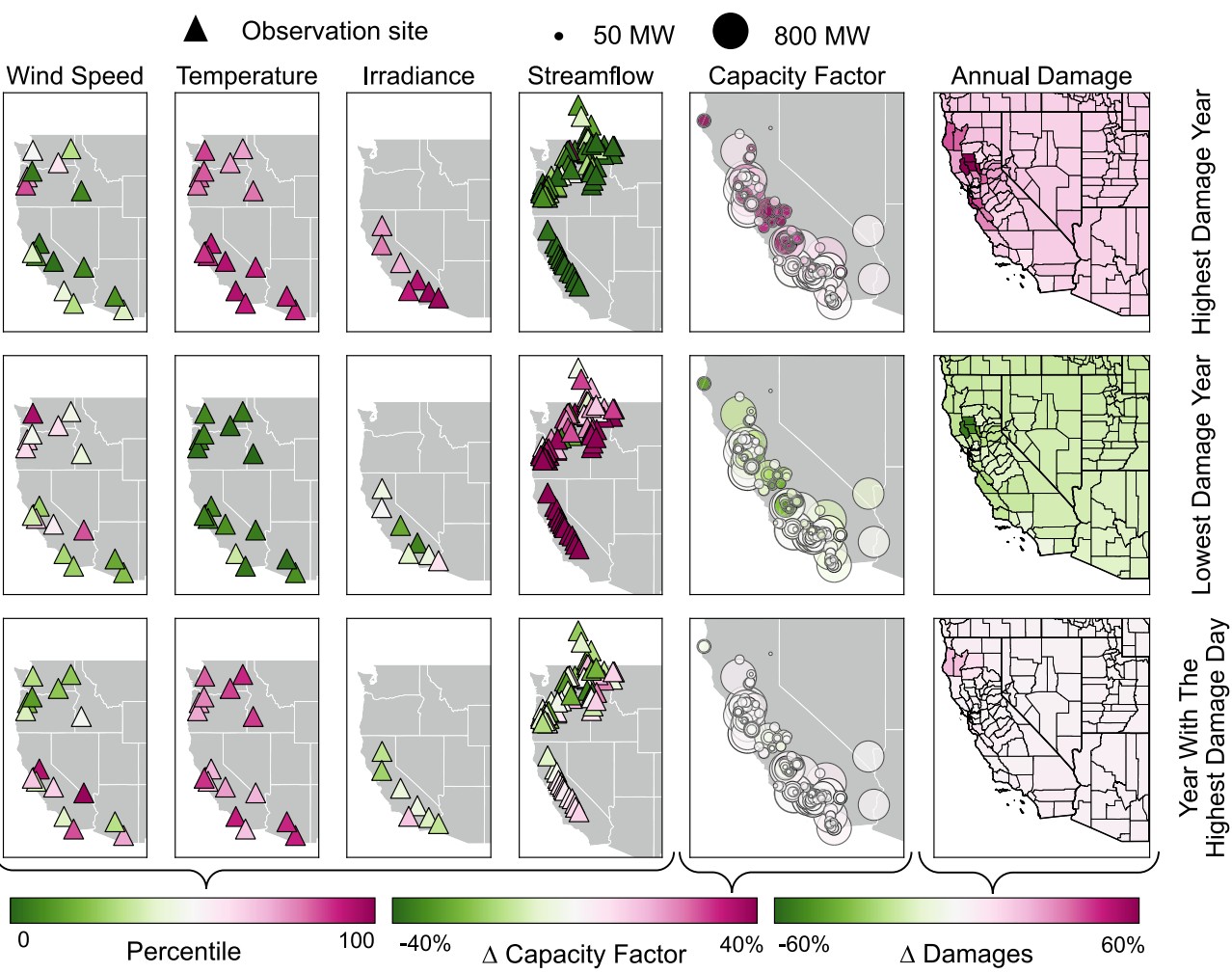

**Fig. 2 | Annual state variables and performance metrics for three distinct years (rows) selected from the 500-year stochastic ensemble.** Top row: the year with the highest air pollution damages; middle row: the year with the lowest air pollution damages; bottom row: the year containing the day with the single highest air pollution damages. Columns 1–4 show hydrometeorological state variables, and columns 5–6 show anomalies in fossil fuel power plants' capacity factors and county level air quality damages. While the years with the highest/lowest damages are clearly associated with extremes in annual streamflow and air temperatures, the year with the highest damage day appears unremarkable apart from elevated temperatures.

during traditional snowmelt months (April–July) before falling to low levels in late summer and fall, while electricity demand peaks during the hottest months (July and August). Figure 1c, d shows corresponding seasonality and uncertainty in daily grid performance, measured in terms of CAISO market prices and health damages from power plant emissions of $SO_2$, $NO_X$, and $PM_{2.5}$, which we refer to in the remainder of this paper as "local" air pollutants (as opposed to carbon dioxide ($CO_2$), a global pollutant). The distribution of daily market prices is heavily skewed during late summer (July–September) when the combination of high cooling demands and a normal seasonal decline in streamflows (hydropower) creates scarcity in the wholesale electricity market. During periods of extreme scarcity, when the CAPOW model cannot feasibly meet electricity demand using available supply, the model activates "slack" variables whose marginal costs of $1000/MWh are set equal to the maximum price permitted in CAISO (according to rules 212 and 214 of the Rules of Practice and Procedure of the Federal Energy Regulatory Commission)[14]. The distribution of daily air pollution damages is less skewed than that of market prices, and we can see the effects of interannual variability in California snow pack and streamflows in Fig. 1d. During wet years, California's peak snowmelt months (May and June) can exhibit very low air pollution damages due to an abundance of spring snowmelt and hydropower, allowing the system operator to avoid using fossil fuels; in dry years, these

conditions are reversed. Like market prices, daily air pollution damages are generally highest from July to September because the system operator must rely more on fossil fuel generators to meet demand due to high demand and declining hydropower availability. The histograms in the bottom row (Fig. 1e–h) show annual average conditions over the 500-year ensemble. Note that the range of average annual prices (30 to 46 $/MWh) is an order of magnitude smaller than the range of daily market prices, which reaches as high as $750/MWh. Similar information for wind and solar power production is available in the Supplemental Information (Supplementary Fig. S3).

**Geospatial visualization of state variables and performance metrics**

Figure 2 further explores how hydrometeorological conditions in California and across the West Coast grid influence power plant operations and air pollution damages at a county level. Each row displays average system conditions for a single year selected from the 500-year stochastic ensemble. Three distinct years are shown: (top) the year associated with the highest total air pollution damages; (middle) the year associated with the lowest total air pollution damages; and (bottom) the year containing the day experiencing the single highest damages across the 500-year stochastic ensemble. The first four columns show the locations of the observation sites used to

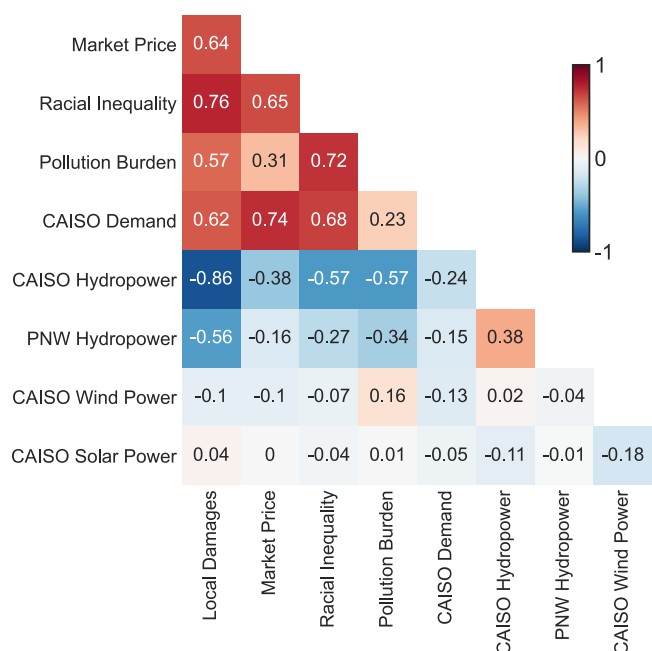

**Fig. 3 | Pearson R correlations between annual system performance metrics and annual state variables.** The cross-sectional values show the correlations between each (row, column) pair. For example, this figure shows that annual local air damages are positively correlated to electricity demand ($R = 0.62$) and negatively correlated to available hydropower in CAISO ($R = -0.86$) and the PNW ($R = -0.56$).

represent wind speed, air temperature, solar irradiance, and streamflow conditions. Triangles are colored based on the annual average percentile at each observation site. The fifth column shows the location of each CAISO power plant in the power systems model. The size of each circle corresponds to power plant size (net installed capacity in MW), while color corresponds to anomalies in each plant's annual capacity factor (a fractional measure of plant usage). For a given year and power plant, anomalies in annual capacity factor are defined as the deviation from each power plant's average capacity factor over the 500-year stochastic ensemble. The last column shows anomalies in air pollution damages for each county in California, calculated in the same manner.

The highest damage year (top row) is among the hottest in the 500-year ensemble. It is also marked by high solar irradiation, relatively low wind speeds, and very low streamflow at almost all measurement gauges. These conditions translate to high demand (especially during summer), high solar power production, and low hydropower production. Note that conditions are also dry in the Pacific Northwest, which compounds scarcity conditions in CAISO by reducing access to imported hydropower. The combination of high demand and low hydropower availability forces the system operator to rely more on fossil fuel power plants, resulting in higher capacity factors. Higher emissions of $SO_2$, $NO_X$, and $PM_{2.5}$ cause increased air pollution damages, with the highest percentage increases in damages occurring around the San Francisco Bay Area and in California's Central Valley.

In the lowest damage year (middle row), conditions are largely reversed: air temperatures and solar irradiance are among the lowest observed over the 500-year ensemble, and streamflows are among the highest observed. The combination of low electricity demand and abundant hydropower allows the system operator to rely less on fossil fuel power plants, resulting in lower air pollution damages.

The third and last row of Fig. 2 shows conditions during the year containing the day with the single highest damages over the 500-year stochastic ensemble. Annual temperatures are elevated, but streamflows

**Table 1 | The summary of the annual average damages of 500-year stochastic ensemble**

| | Annual average damages | Per capita | Per capita (people of color population) | Per capita (White population) |
|---|---|---|---|---|
| No tax | $ 840,818,811 | $ 21.27 | $ 22.46 | $ 19.56 |
| Local tax | $ 260,675,737 | $ 6.59 | $ 6.60 | $ 6.58 |
| **Percent reduction** | **69%** | **69%** | **71%** | **66%** |

in California are only slightly above the median. A north-south dipole in wind speeds and streamflows is apparent with lower values in the PNW. Apart from the elevated temperatures, this year appears unremarkable from a hydrometeorological standpoint. Capacity factors at fossil fuel power plants in CAISO are close to normal, as are air pollution damages.

### Relationships between hydrometeorology and performance metrics

Figure 3 shows correlations among annual system states and performance metrics for the 500-year stochastic ensemble simulation ($p < 0.05$ except for some correlations involving solar and wind power). In addition to health damages and market prices, socio-economic and environmental outcomes now include two measures of equity: (1) racial inequality, defined here as the correlation between counties' air pollution damages and percentage of residents who identify as "people of color"; and (2) pollution burden inequality, defined here as the correlation between counties' air pollution damages and the average CalEnviro Screen Score of the Census Tracts contained within the county, which measures relative pollution burdens from all pollution sources[15].

Keeping in mind that these correlations are based on 500 years of annual data, we find that hydrologic conditions are the most important driver of human exposure to power plant emissions, with drought increasing exposure. Due to CAISO's reliance on imported hydropower from the PNW, we also find significant negative correlations between PNW hydropower availability and air pollution damages in California. Dry years are also correlated with years with high electricity prices. Electricity demand in the CAISO market—largely driven by cooling demands—also shows strong positive correlations with market prices and air pollution damages. Simultaneous hot-dry (high demand, low hydropower) years result in the worst outcomes (the highest damage year shown in the top row of Fig. 2 is an example).

Our results also show that both drought (reduced hydropower production) and elevated temperatures (higher electricity demand) exacerbate air pollution inequalities. An overwhelming majority of California's power plants are located in counties with mostly (>50%) people of color, and those counties receive most of the air pollution damages created by power plants in the state. Average annual air quality damages over the 500-year stochastic weather ensemble are $22.46 per capita for people of color and $19.56 per capita for White residents (see Table 1). Figure 3 shows that in high demand/low hydro (i.e. hot/dry) years, the resultant increased air pollution from power plants disproportionally impacts counties with majority people of color populations, and counties that already experience a higher burden from other pollution sources. Figure S4 in the Supplemental Information and the accompanying discussion provides additional information about how drought and extreme heat exacerbate existing pollution inequalities.

Note that our results fail to reveal strong correlations between annual wind and solar power availability and performance metrics. This result, which confirms previously reported findings[6,16], is due to the relatively small percentages of California electricity demand that is met by wind (4.5%) and solar (10.7%) in the 2018 version of the grid, as well as the smaller range in interannual variability observed in wind

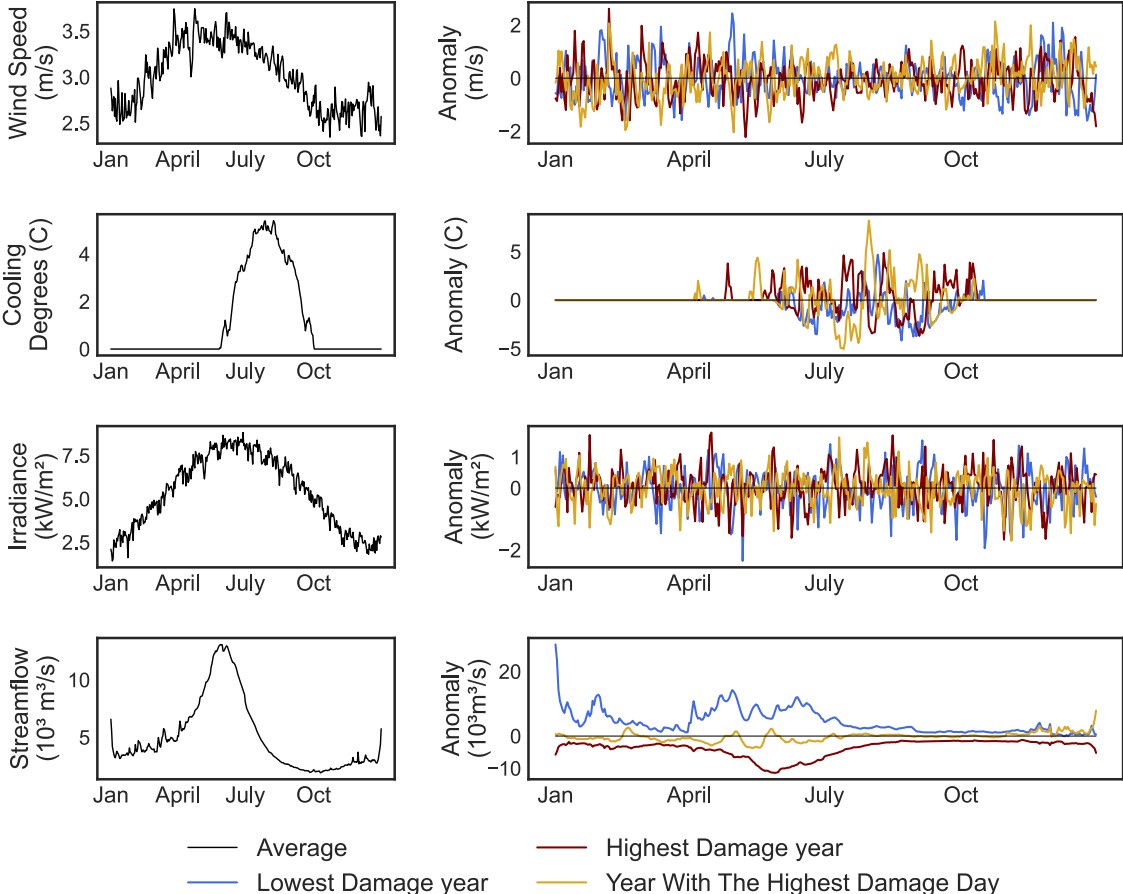

Average   —— Highest Damage year
—— Lowest Damage year   —— Year With The Highest Damage Day

**Fig. 4 | Daily anomalies in hydrometeorological conditions for the same three years that are featured in Fig. 2.** The first column shows the average 365-day profiles of wind speed (first row), cooling degree days (second row), solar irradiance (third row), and streamflow (fourth row). The second column shows daily anomalies relative to the average profile for the highest damage year (red line), lowest damage day (blue line), and the year with the year with the highest damage day (gold). The prominent, positive cooling degree day anomaly shown for the gold line (the year with the highest damage day) represents a summer heat wave occurring around July 30th, which significantly increases demand. At this time of the year, hydropower availability is generally low, and the heat wave forces the system to use all other available resources, including high polluting power plants. This results in a day with extremely high air pollution damages.

speeds and solar irradiance compared to hydropower (see Supplementary Fig. S3).

A similar figure showing correlations calculated from daily data over the 500-year ensemble is provided in the Supplemental Information (Fig. S5), though daily correlations do not include equity measures since those are determined on an annual basis. Electricity demand is the strongest driver of emissions damages on a daily time step. We also find that the influence of variable renewable energy availability on modeled outcomes is greater on a daily time step.

Figure 4 examines daily anomalies in hydrometeorological variables for the same three years featured in Fig. 2 (the highest damage year, the lowest damage year, and the year with the worst day). The left column of panels in Fig. 4 shows average 365-day profiles of wind speeds, cooling degree days (positive anomalies in temperature above 18.33 °C), solar irradiance, and streamflow, calculated by averaging across all observation sites and all years in the stochastic ensemble. The right panels track daily anomalies relative to the average profiles at left for the highest damage year (red), the lowest damage year (blue), and the year with the highest damage day (gold). This comparison helps illustrate how different hydrometeorological events can underpin chronic (seasonal, annual) versus acute (daily) increases in air pollution damages from power plants.

In the year containing the highest damage day (gold line), a severe, late summer heat wave (indicated by a large positive anomaly in cooling degree days) occurs on July 30th. This heat wave causes a spike in electricity demand, leading to significant increases in electricity production at fossil fuel power plants and human health damages from local air pollutants.

Note that the year with the highest damage day experiences close to average hydrologic conditions throughout the entire year. However, the spike in electricity demand on July 30th does coincide with limited hydropower availability, which is low in late July even under *normal* hydrologic conditions. Faced with extremely high electricity demand and limited hydropower, the system operator makes use of every available fossil fuel power plant to produce electricity, which causes air pollution damages to reach their highest levels across the 500-year ensemble. This exact day can also be observed in Fig. 1c and 1d as the highest observed peaks in market prices and air pollution damages.

The CAISO system is thus capable of experiencing supply short-falls even in years of normal hydropower availability. Confirming the real experience of CAISO system operators in recent years[17], we find that *late summer heat waves*, lasting days to weeks, pose acute operational risks because, in large part, they coincide with normal seasonal declines in snowmelt and hydropower production. In contrast, the role of anomalous hydrological conditions (i.e. drought) in driving system performance is generally more chronic in nature. Figure 4 shows that streamflow anomalies during the (dry) highest damage year are most pronounced from December to July. Thus, when drought affects the CAISO system, corresponding reductions in

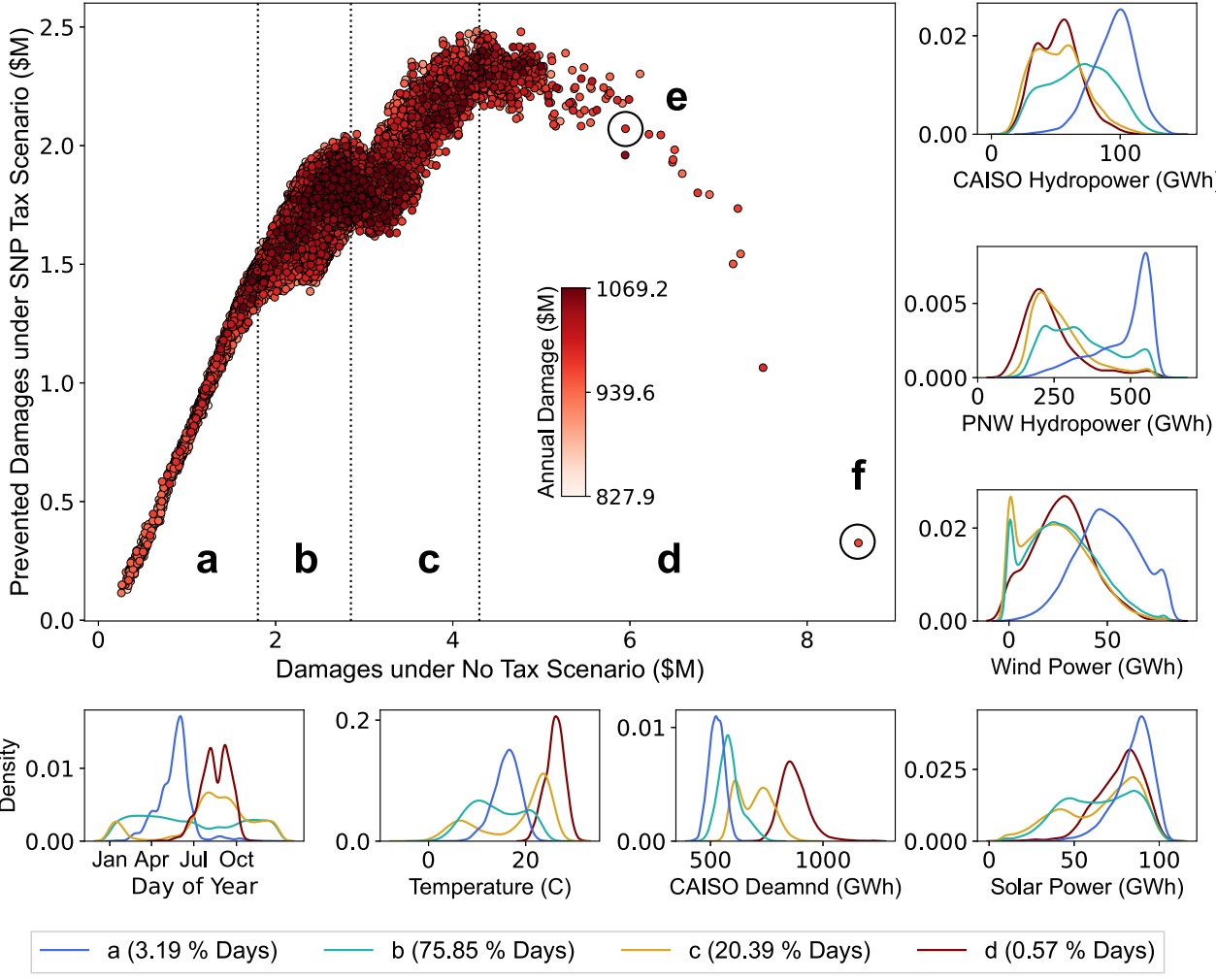

**Fig. 5 | The large panel shows all simulated days' base case damages (x-axis) vs. damages prevented by local air tax (y-axis).** The dots are colored based on their corresponding year's annual base case (no tax) damages. The surrounding plots show distributions of system conditions associated with each of the large panel's four main zones ("a" through "d"). Prevention of damages under a local tax increases with base case damages, up to a point. In Zone "d" (maroon line in distribution plots) the local tax loses its efficacy. These are very hot, late summer days that occur when hydropower availability is seasonally low. Consequently, the system operator must use every power plant at its disposal to meet the demand, so the local tax cannot prevent the damages.

hydropower availability and increases in air pollution damages can occur over this entire several-month period.

## Weather conditions impact emissions control policy effectiveness

We simulated human health damages in California from exposure to air pollutant emissions under four different policy scenarios. Each policy tested varied the dollar per megawatt hour penalty placed on emissions from each generator in the CAISO system: (1) a base case scenario in which no emissions penalties are enacted; (2) a "local tax" scenario in which penalties are placed on emissions of local pollutants only ($PM_{2.5}$, $SO_2$, and $NO_x$); (3) penalties on $CO_2$ equivalent emissions only; and (4) penalties on both local and $CO_2$ emissions. Penalties were only applied to generators in CAISO. We did not penalize generators in the Pacific Northwest in order to focus on CAISO and reduce the potential for confounding variables that could affect our results. Penalties for each individual generator were set equal to the dollar per megawatt hour rates estimated by the AP3[18] integrated assessment model (for $PM_{2.5}$, $SO_2$, $NO_x$) and EPA's Social Cost of Carbon (SCC) ($47.38 per metric ton in 2018 for $CO_2$) and added to each generator's marginal cost of power generation. All four policy scenarios were run under the same 500-year stochastic weather ensemble. Results for

scenarios (3) and (4) involving power plant $CO_2$ emissions are available in the SI but are not a focus of our discussion, because $CO_2$ penalties generally do not change the "merit order" of the CAISO market's supply curve or hourly system operations (see Supplementary Fig. S8 and section on influence of penalties on power plant emissions).

Table 1 provides a summary of damages measured for the base case (no tax) and local air tax. For the base case, average annual damages over the 500-year stochastic ensemble are about $840 million ($21.27 per capita), and the local tax reduces this number to $260 million ($6.59 per capita), about a 69% reduction in damages (see Supplementary Fig. S9). Damages per capita for the people of color are $22.46 under the base case, and a local tax decreases that to $6.60 (a 71% decrease). Damages per capita for the White population is $19.56 under the base scenario and $6.58 under the local tax (a 66% decrease).

Figure 5 explores how hydrometeorological conditions influence the effectiveness of the local tax at reducing health damages. The large panel shows information for each day in the 500-year stochastic ensemble (1 dot = 1 day). The x-axis measures air pollution damages on a given day under the base case (i.e., without any emissions penalty in place), while the y-axis measures damages *prevented* on that same day with the local tax in place (the difference in damages between the base and local tax scenarios for that day). The dots are colored by total

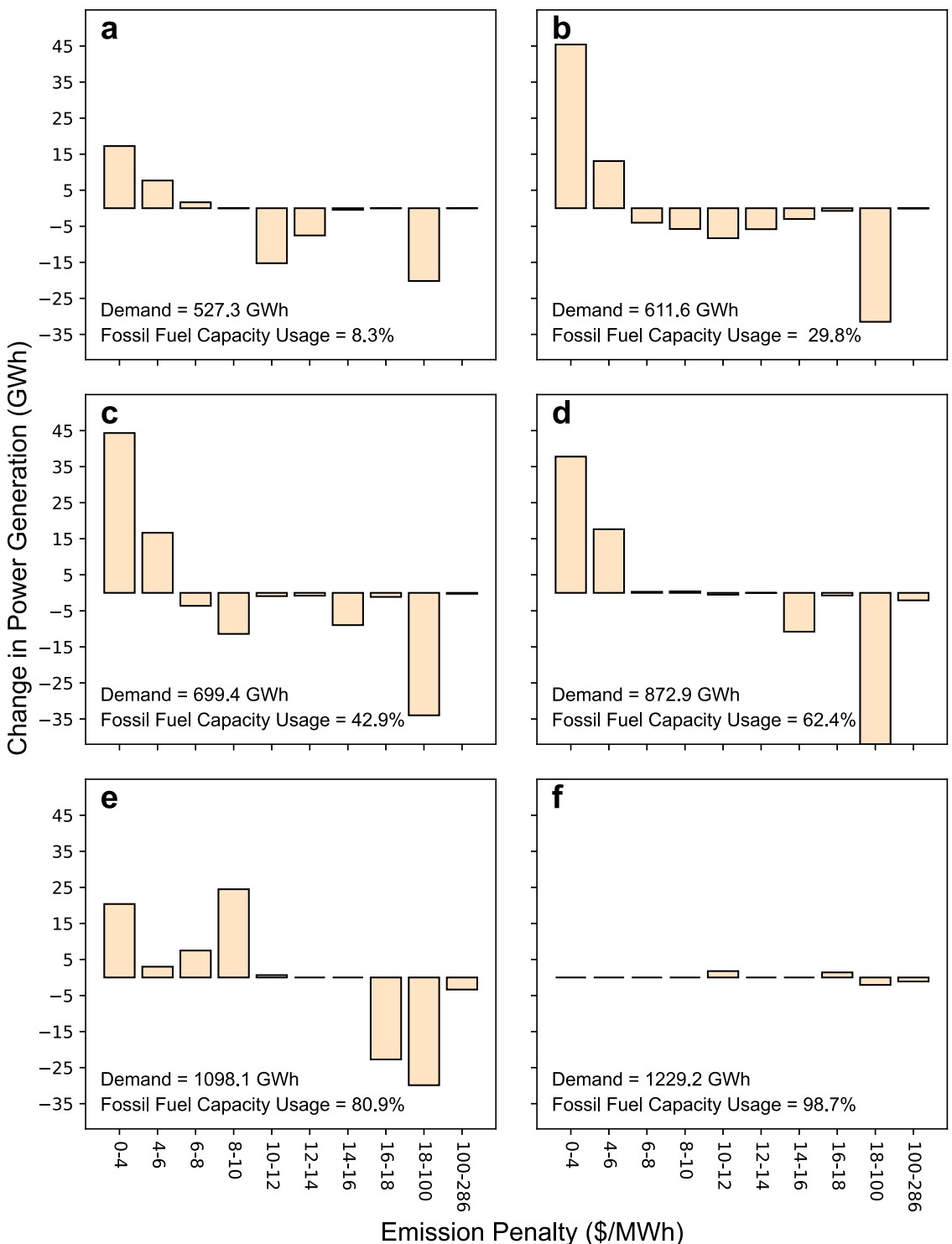

**Fig. 6 | The effect of the local tax on generator usage for each zone of Fig. 5.** Generators are binned based on their local tax penalty (*x*-axis), and the *y*-axis shows the change in each generator group's average generation with local tax relative to the base scenario. The numbers on the bottom left corner of each panel show the corresponding zone's daily average electricity demand and the percentage of fossil fuel capacity usage. The local tax shifts the power generation from more harmful generators (higher $/MWh damages) to less harmful generators. However, in panel (**f**) almost 99% of generation capacity is exhausted, and the policy intervention is not able to alter operations relative to the base case.

annual air pollution damages under the base case for the specific year in which they fall.

The cloud of points in Fig. 5 has been divided into four zones. Zone "a" (containing 3.2% of all days) shows a positive linear relationship between base case (no tax) damages and prevented damages under the local tax scenario. In other words, the more damaging a day

would be, the more damages avoided with the emissions control policy in place. The distribution plots surrounding the main panel indicate that days in Zone "a" are concentrated during spring snowmelt (April-July) when hydropower availability in CAISO and the PNW is highest. Days in Zone "a" are also marked by mild temperatures, low electricity demand, and high availability of both wind and solar power. These are

conditions that make it *easy* for the system operator to meet electricity demand without relying significantly on fossil fuel generators whose emissions of $SO_2$, $NO_X$, and $PM_{2.5}$ are associated with higher health damages.

Zone "b" contains 75.9% of all days. We again observe a positive (though somewhat weaker) relationship between base case damages and damages prevented under the local tax scenario, indicating that it remains feasible and cost effective for the system operator to shift its reliance towards less damaging fossil fuel power plants. The distribution plots surrounding the main panel show that Zone "b" days can occur anytime during the year, and they are diverse in their underlying characteristics, generally exhibiting lower demand but a wide range of possible wind, solar, and hydropower availabilities. Days in Zone "c" (containing 20.4% of all days) are concentrated in late summer/early fall (July through October) when hydropower availability is low, and demand is high. On these days, the system operator must increase its use of fossil fuel power plants in order to meet demand, but it is largely able to avoid the use of the most damaging (and heavily penalized) plants.

In Zone "d" (0.6% of all simulated days), we see a sharp drop in prevented damages even on days with extremely high base case damages. On these days, the local air pollution tax breaks down in effectiveness. Zone "d" days share a similar profile with Zone "c" days in terms of day-of-year (late summer/early fall), the low availability of hydropower, and wind and solar generation potential. The feature that distinguishes Zone "d" from Zone "c" days is air temperatures. Zone "d" days are almost always associated with heat waves (extremely high electricity demand), which limits the ability of the system operator to shift its reliance away from the most damaging power plants. Note in particular that the coloring of dots (days) falling within Zone "d" ranges widely. Most (76%) of Zone "d" days occur in years with base case damages above the 50th percentile (see Supplementary Fig. S9), and these higher damage years tend to be dry. However, our results demonstrate that drought is not a necessary pre-condition for failure of the power grid (i.e. the occurrence of rolling blackouts), nor is it necessary for the local air tax to (temporarily) lose the ability to reduce pollution damages. The heat waves that lead to these extreme conditions can occur even in "normal" hydrologic years, but are most problematic if they occur during post-snowmelt (summer and fall) months that are characterized by seasonally low hydropower availability.

Figure 6 provides a more detailed look at the supply side dynamics at work on days in zones "a" through "d," as well as on two individual days within Zone "d," labeled "e" and "f." Along the *x*-axis, electricity production is aggregated and binned based on the human health damages associated with each generator, i.e., the penalty applied to each generator under the local tax scenario. The y-axis tracks the change in the generation due to the local tax, relative to the base case. Each panel also contains information about average demand conditions and the percentage of fossil fuel plant capacity being utilized in each zone.

The local tax is designed to incentivize the system operator to shift its reliance away from more damaging and more heavily penalized power plants towards those causing lower air pollution. Figure 6 confirms that this indeed occurs almost all of the time. In zones "a" through "c" (cumulatively representing 99.9% of all simulated days), generators with emissions damages between 0 and 4 $/MWh are favored, and the system decreases its reliance on more heavily penalized generators. Zone "d" days experience significantly higher demand on average, and the system operator must rely on more damaging power plants (6–10 $/MWh), while still largely avoiding the use of the most harmful plants. Figure 6e shows information for a single day ("e" in Fig. 5) that experiences even higher demand; utilization of total fossil fuel power capacity jumps to 81%, necessitating greater use of power plants with damages of 6–10 $/MWh. Still, the system operator can avoid the use of the most harmful plants. In

Fig. 6f, however, we observe that the emissions control policy is no longer able to incentivize any changes in generation between the base case and local tax scenarios. On this day, indicated as point "f" in Fig. 5, extremely high demand requires the system operator to make use of 98.7% of its fossil fuel capacity throughout the day. In some hours, electricity demand exhausts the system's capacity, resulting in a loss of reliability (blackouts) and market prices of $1000/MWh. This point is the single worst day in the 500-year ensemble and is also observable in Fig. 1 and Fig. 4 (bottom row) (the latter shows the air temperature anomaly that causes the rolling blackout, extreme high prices, and high air pollution). Supplementary Figures S6 and S7 show a similar set of correlation matrices to Fig. 3 and Supplementary Fig. S5, describing dependencies among weather-based state variables and performance metrics under the local tax scenario.

Note that in our simulations we do not institute an explicit pollution limit or target emissions reduction. Instead, we incorporate negative externalities related to air pollution into the marginal cost of electricity production. By doing so, it almost always alters the minimum cost solution of the CAPOW model (heavier polluting power plants are used less in order to minimize costs). In a few instances (e.g. the "worst day" shown above in Fig. 5 and Fig. 6f) we observe that the optimal power plant operating schedules with and without the pollution tax are nearly identical. When this occurs, the system operator (represented as a mathematical program) is rationally avoiding loss of load (in our model valued at $1000/MWh) instead of avoiding air pollution damages (>>$1000/MWh).

## Discussion

The main goal of this study is to quantify the role of hydrometeorological uncertainty and extremes, especially droughts and heat waves, in driving human exposure to power plant emissions of $SO_2$, $NO_X$, and $PM_{2.5}$. Focusing our analysis on the U.S. West Coast and California, we simulated hourly grid operations across an ensemble of 500 synthetic years representing stationary weather uncertainty and translated emissions from individual power plants to estimates of human health damages on a county level. The first part of our analysis used outputs from this 500-year simulation to identify the key hydrometeorological conditions that underpin "good/bad" years marked by low/high human health damages from power plant emissions, as well as the conditions that cause shorter term spikes in emissions ("bad" days). We also tracked two measures of inequality to better understand how increases in power plant emissions impact different communities. The second part of our analysis explored how hydrometeorological conditions, especially extremes, influence the effectiveness of a tax on local air pollutants, in which individual generators are financially penalized on a $/MWh rate based on their emissions of $SO_2$, $NO_X$, and $PM_{2.5}$.

A few limitations and modeling assumptions should be acknowledged when interpreting our findings. First, computational limitations obligated us to simplify the grid operations model in some aspects, most notably the network topology of the grid, which assumes that transmission line constraints exist among but not within aggregated model zones (see Supplementary Fig. S2). We also do not consider planned or forced outages of generators or transmission lines, meaning we very likely underestimate the probability of extreme scarcity conditions affecting the grid, and thus also the probability that an emissions control policy like the local tax will not adequately incentivize a reduction in emissions. In addition, our calculation of $/MWh human health damage rates for each power plant is based on the AP3 integrated assessment model, which measures the effects of annual average exposure to emissions. We thus assume constant damage rates at each plant throughout the year and across years. In reality, excessively high temperatures accelerate the air chemistry reactions that produce ground level ozone[19–21], while prolonged precipitation deficits can increase the production of dust and other aerosols[22,23] and

prevent the natural washout of vapor phase and particulate bound chemicals via wet deposition. Not accounting for these dynamics may lead us to underestimate the severity of high damage days and over-estimate the severity of low damage days. Moreover, this study only examines the 2018 version of the CAISO and larger West Coast power grids operating under stationary hydrometeorological uncertainty. Future studies should consider planned decarbonization efforts in West Coast states and/or the effects of climate change[24].

Based on our 500-year simulations, we find that hydrologic conditions in California, and to a lesser extent, the PNW, are the primary driver of good/bad years as compared to fluctuations in load, wind and solar. This is likely to persist in the future even as wind and solar and capacity grow, due to larger interannual variability in hydrologic conditions compared to wind speeds and solar irradiance[16]. Dry years are associated with low hydropower production, leading to more harmful power plant emissions from fossil fuel generators and also contributing to higher market prices, which ultimately increase consumer costs. We also find that dry years exacerbate inequalities in terms of which communities are harmed by power plant emissions, with drought-caused increases in emissions disproportionately impacting counties with a higher % of people of color and counties with greater shares of the pollution burden from all sources.

However, annual hydrologic conditions are less useful at predicting whether a year will contain individual days with extreme levels of air pollution from power plants. A key finding from this study is that heat waves occurring in late summer and early fall are the primary cause of days with extremely high air pollution damages. These events cause significant short-term increases in electricity demand during a time of the year when water and hydropower availability is limited even in "normal" hydrologic years. In response, the system operator must deploy nearly every fossil fuel power plant available in order to avoid instituting rolling blackouts, leading to very high pollutant emissions and human health damages, as well as extremely high market prices.

In addition, our results suggest that while instituting financial penalties on power plant emissions of $SO_2$, $NO_X$, and $PM_{2.5}$ is highly effective at curtailing human health damages, with reductions relative to baseline occurring in over 99% of days, this type of emissions control policy does not adequately incentivize emissions reductions during periods of extreme scarcity on the grid. Again, we find that late summer heat waves are the most common cause of these extreme conditions.

The results of our computational experiments could affect how grid participants including system operators and policy makers plan around hydrometeorological extremes and design mechanisms to protect humans from the combined effects of excessive heat and air pollution damages. For example, where possible, strategies for managing grid scarcity that involve curtailment of electricity service should take into account the economic cost of (and inequalities in) air pollution exposure. Voluntary programs that compensate consumers for reducing electricity usage on critical days should incorporate the cost of air quality damages (thereby increasing the incentive to comply). Involuntary load reduction strategies (i.e. rolling blackouts) should avoid targeting communities and times-of-day that experience the highest air pollution exposure, in order to allow residents in the most affected areas to remain indoors during periods of acute stress. Adaptive strategies for managing heat stress (e.g. cooling centers) should take into account risks for electricity supply on very hot days.

Future work should incorporate dynamic estimates of human health damages from power plant emissions, especially estimates that take into account the influence of weather conditions on air chemistry reactions. Our current model uses a static estimation of human health damages, even though it is generally known that the same emissions on one day can be more/less harmful depending on weather conditions. Incorporating a dynamic estimation could give us more realistic estimates of health damages to inform better planning. In addition,

this model uses a relatively simple five-zone representation of the West Coast power grid. Another area of future work could be to use a more detailed power grid model, which could address the role of higher resolution grid dynamics (e.g. transmission congestion) on system outcomes. Although there is already a body of research demonstrating air pollution reduction as a significant co-benefit of decarbonization[25–28] (including reduced inequality[29] in damages from exposure), an open question is how drought and heat waves will affect air pollution damages in expanding and decarbonizing electricity systems.

## Methods

We use the California and West Coast Power System (CAPOW) model to simulate the hourly operation of California's bulk electric power system and wholesale electricity market. CAPOW's spatial domain covers most of the larger West Coast grid, including transmission pathways linking California with the Pacific Northwest and Southwest, which California uses to import significant amounts of electricity (see Supplementary Fig. S2). Within California, the model covers the operations of the California Independent System Operator (CAISO), which manages approximately 80% of California's electricity flow. CAPOW is a Python-based open-source simulation framework specifically designed for evaluating risks from hydrometeorological uncertainty and extremes in bulk power systems and wholesale electricity markets. The model has been validated[5,6,13] and used previously to explore dynamics of the West Coast grid under long term climate and technological change[16,30]. A stable version of the code and relevant data are freely available via online public repositories[31].

CAPOW has two core components: (1) power system dispatch models of the West Coast bulk electric power system that make use of an aggregated, zonal topology; and (2) a "stochastic engine" that generates synthetic time series records of spatially distributed hydrometeorological variables (streamflow, air temperatures, wind speeds, and solar irradiance) and translates these to relevant power system inputs (daily volumes of available hydropower, hourly electricity demand, hourly solar power production, and hourly wind power production). The following sections provide details about these two core model components.

### Power systems model

The power systems model in CAPOW employs a simplified network topology made up of five major load zones (four in California and one in the Pacific Northwest) that are linked by aggregated high-voltage transmission pathways. Each zone is associated with its own portfolio of generating resources taken directly from the 2018 U.S. EPA eGrid database, as well as separate hourly time series of electricity demand, daily hydropower availability, and hourly wind and solar production. Power system operations are simulated using two separate unit commitment economic dispatch (UC/ED) formulations, one that simulates the operation of the CAISO wholesale electricity market in California and one that simulates the operation of the informal Mid-Columbia (Mid-C) market in the Pacific Northwest. Dynamic exchanges of electricity between California and neighboring regions like the Pacific Northwest and Southwest are modeled statistically (see the section on synthetic weather generation for more detail) and then treated as a constraint in each UC/ED formulation.

The UC/ED for each market is coded as an iterative, mixed-integer linear program; the objective function of each program is to minimize the cost of meeting fluctuating hourly electricity demand and required operating reserves, subject to additional constraints on meeting electrical interchange demands, the capacity of transmission pathways linking the zones within the CAISO market, and operating limits on individual generators. Given input time series of hourly electricity demand, available solar and wind power generation, and available daily hydropower generation in each zone, the optimization routines

schedule generation at dispatchable power plants according to each market's least cost objective, employing a forward-looking operating horizon of 48 h. Note that the CAPOW model does not account for random or correlated (e.g., weather dependent) forced power plant de-ratings and outages, nor does it represent the use of forecasts or the operations of balancing/real-time electricity markets. These assumptions (and likewise, CAPOW's simplified representation of the West Coast grid's transmission system) are made to reduce computational run-time and facilitate uncertainty characterization. However, they represent a source of bias in our results that likely underestimates the probability of reliability failures (rolling blackouts). This does have relevance to our results, which show a very low likelihood of grid failures, but a strong connection between days with resource adequacy issues and high levels of air pollution from power plants. Extreme days during which very high prices and very high air pollutant emissions co-occur while grid reliability simultaneously falters may be underrepresented by our model. It is also important to note that in this study, we do not consider the impacts of future climate change on underlying distributions of hydrometeorological variables as we choose instead to examine system performance under "stationary" uncertainty (see the section on synthetic weather generation for more detail).

Performance metrics tracked by CAPOW include individual generators' power production (MWh) and the hourly wholesale electricity price ($/MWh). The market price of electricity is calculated hourly for each zone as the dual (shadow price) of each zonal energy balance constraint. The weighted average price across all zones is used to determine the overall ('hub') price, with weights obtained via a regression trained on historical (2012-2016) price data in the CAISO market. We followed the same procedure as Holland et al.[32] to determine air pollution damages (measured in dollars) from all generators in the CAISO footprint. Given the time series of electricity production at each generator, carbon emissions are estimated using the EPA reported average heat rate and the fuel carbon intensity. Global damages per unit of $CO_2$ emitted are valued at the EPA's Social Cost of Carbon (SCC) ($47.38 per metric ton in 2018) for all generators. Emissions of local pollutants ($PM_{2.5}$, $SO_2$, and $NO_X$) were estimated using time series of modeled electricity production at each generator along with EPA reported pollution rates on a metric ton per MWh basis. Damages caused by $PM_{2.5}$, $SO_2$, and $NO_X$ emissions were calculated using separate dollar per MWh ($/MWh) rates for each individual generator estimated by the AP3[18] integrated assessment model. AP3 models the physical dispersion of primary $PM_{2.5}$, $SO_2$, and $NO_X$ emissions and chemical processes in the atmosphere (for $SO_2$ and $NO_X$ emissions) to translate modeled emissions from each generator's smokestack to ambient concentrations of $PM_{2.5}$ at counties in the contiguous United States. Then it translates county level $PM_{2.5}$ concentrations to premature mortality risks using concentration-response functions. Finally, it monetizes mortality risk using the value of statistical life[33]. Local air pollutant damage calculations are based on 2017 population levels, and they assume static and uniform air chemistry interactions, meaning modeled damage rates do not change sub-annually or inter-annually. It is important to note that we assume that damages caused by individual power plants do not change on a day to day or year to year basis.

We used data from CalEnviroScreen 4.0[15] to estimate measures of inequality, focusing on: (1) racial inequality, calculated as the correlation between county level power plant damages and the fraction of each county's population that is people of color; and (2) pollution burden inequality, calculated as the correlation between county level power plant damages and each county's CalEnviro screen score.

## Stochastic engine
Although the long-term impacts of climate change could significantly influence the cost and reliability of the West Coast grid, we do not

consider these effects in this study, choosing instead to focus on operational risks facing the current (2018) version of the system under stationary weather uncertainty. Yet, relying exclusively on historical hydrometeorological observations would critically bias our vulnerability assessment, especially concerning risks from compound extremes (e.g., defined here as multiple concurring extremes in correlated exogenous forcings and/or endogenous system failures). In order to capture a greater number of these events, we first use a synthetic weather and streamflow generator to create a 500-year ensemble of daily hydrometeorological conditions (streamflows, air temperatures, wind speeds, and solar irradiance) at observation sites across the West Coast. These data are then translated to time series of relevant power system inputs (daily hydropower production, hourly demand, hourly solar power availability, and hourly wind power availability) for each UC/ED[34–38]. Please see Supplemental Information section labeled "Additional explanation of CAPOW" for a detailed description of data inputs, products, and modeling approaches used by the stochastic engine.

Several historical datasets form the basis of CAPOW's synthetic weather generator. These include daily average wind speed and air temperatures from 17 major airports across the west coast (Fig. 2)[39], taken from the NOAA Global Historical Climatological Network. Data cover the period 1970–2017 for air temperatures and 1998–2017 for wind speeds. The missing wind speed data at each site (1970–1998) are replaced via a bootstrapping procedure that uses daily average temperatures and day-of-year to condition selection of historical wind speed data from 1998–2017 to substitute. The records of global horizontal irradiance at six sites for the same period are taken from National Renewable Energy Laboratory's National Solar Radiation Database (NSRDB)[40]. Recorded daily streamflows over the 1954-2008 period for 108 sites across the Pacific Northwest and California are taken from the BPA Modified Streamflow database[41] and the California Data Exchange Center (CDEC)[42].

The synthetic weather generator creates ensembles of hydrometeorological data that maintain the same dependencies (spatio-temporal dynamics and cross-correlations among variables on annual, seasonal, daily, and hourly time scales) and statistical moments as the historical record. First, daily average wind speed and temperature data were used to create 365-day average profiles for each site. The irradiance data was used to create an average 365-day "clear sky" profile for each site:

$$\mathbf{TP_n} = \frac{1}{Y}\sum_{y=1}^{Y}\mathbf{T_{n,y}} \tag{1}$$

$$\mathbf{WP_n} = \frac{1}{Y}\sum_{y=1}^{Y}\mathbf{WS_{n,y}} \tag{2}$$

$$\mathbf{SP_n} = \frac{1}{Y}\sum_{y=1}^{Y}\mathbf{S_{n,y}} \tag{3}$$

where, $\mathbf{TP_n}$ = average temperature on calendar day $n$ across $Y$ years (°C), $\mathbf{T_{n,y}}$ = observed temperature on calendar day $n$ in year $y$ (°C), $\mathbf{WP_n}$ = average wind speed on day $n$ across $Y$ years (m/s), $\mathbf{WS_{n,y}}$ = observed wind speed on day $n$ in year $y$ (m/s), $\mathbf{SP_n}$ = average clear sky irradiance on day $n$ across $Y$ years (W/m²), $\mathbf{S_{n,y}}$ = observed clear sky irradiance on day $n$ in year $y$ (W/m²).

Then "anomalies" (deviations from average temperatures and wind speeds) and "losses" in irradiance from cloud coverage were generated by subtracting the 365-day profile from observed data over the period 1998-2017, the longest period with historical observations

available across all sites:

$$RT_d = T_d - TP_n \qquad (4)$$

$$RW_d = WS_d - WP_n \qquad (5)$$

$$IL_d = SP_d - I_n \qquad (6)$$

where, $RT_d$ = residual temperature on day $d$ (°C), $RW_d$ = residual wind speed on day $d$ (m/s), $IL_d$ = irradiance "losses" on day $d$ (W/m²). $I_n$ = Observed irradiance

Residual data were transformed to Gaussian distributions (Eqs. 7–9) and used to fit vector autoregressive (VAR) models that capture daily autocorrelation and covariance across variables (Eq. 10). The Akaike Information Criteria (AIC) was used to determine the number of lags. In simulation mode, the error terms in the VAR model were then stochastically generated from the Gaussian distribution of the covariance matrix of the residual dataset. Stochastically generated weather process residuals were then back-transformed and added to the average profiles, yielding synthetic values of daily irradiance, temperature, and wind speeds at each site:

$$WRT_d = \widehat{RT_D}/\sigma T_n \qquad (7)$$

$$WRW_d = \widehat{RW_D}/\sigma W_n \qquad (8)$$

$$WIL_d = \overline{ILD}/\sigma IL_n \qquad (9)$$

where $WRT_d$ is the whitened residual temperature on day d, $WRW_d$ is the whitened residual wind speed on day d, $WIL_d$ is the whitened irradiance losses on day d, $\widehat{RT_D}$ is the mean shifted, log-transformed residual temperature on day d (°C), $\widehat{RW_D}$ is the mean shifted, log-transformed residual wind speed on day d (m/s), $\widehat{IL_D}$ = is the mean shifted, log-transformed irradiance losses on day d (W/ m²). $\sigma T_n$ is the standard deviation of transformed temperature residuals on calendar day n, $\sigma W_n$ is the standard deviation of transformed wind speed residuals on calendar day n, $\sigma IL_n$ is the standard deviation of transformed irradiance losses on calendar day n

$$y_t = C + A_1 y_{t-1} + A_2 y_{t-2} + \cdots + A_p y_{t-p} + \varepsilon_t \qquad (10)$$

where $y_t$ is the $k \times 1$ vector of simulated values of each variable $C = k \times 1$ vector of constants, $A_i = k \times k$ matrix of coefficients, $\varepsilon_t = \times 1$ vector of error terms, $t$ is the time period, $p$ is the model lag.

For the synthetic streamflow, we used a two-step process in order to capture the statistical dependences of total annual streamflow on air temperatures at an annual and sub-annual time step. At an annual time step, we used Gaussian Copulas to generate annual records of total streamflow and temperatures. We converted longer observed temperature records (1958–2008) at the meteorological stations closest to streamflow gauges into heating and cooling degree days (HDDs and CDDs, respectively), which are daily temperature deviations from 18.33 degrees Celsius. The sum of daily HDDs and CDDs for each calendar year were calculated as the total annual HDDs and CDDs, providing a rough measure of each historical year's "hotness" and "coolness:"

$$HDD_{d,s} = \max(18.33 - T_{d,s}, 0) \qquad (11)$$

$$CDD_{d,s} = \max(T_{d,s} - 18.33, 0) \qquad (12)$$

where $HDD_{d,s}$ is the heating degree days on day $d$ at station $s$, $CDD_{d,s}$ is the cooling degree days on day $d$ at station $s$, $T_{d,s}$ is the average near surface air temperature on day $d$ (°C) at station $s$.

Empirical cumulative probability distributions of total annual HDDs, CDDs, and total streamflow for all sites were transformed into quantile space:

$$P = P(Q \geq q) \qquad (13)$$

where $Q$ is the variable of interest (total annual streamflow, annual HDDs, or annual CDDs at a given site).

To ensure coherent mean-zero data, the empirical distributions were transformed again into uniform distributions between −1 and 1:

$$Y = 2(P - 0.5). \qquad (14)$$

Using values of Y, a multivariate Gaussian distribution was fitted. Then synthetic values of HDDs, CDDs, and annual streamflow and random samples were drawn from it and back-transformed reversing the Eqs. (13) and (14).

Total annual streamflow was then disaggregated to a daily time step using an approach that allows the synthetic streamflow ensembles to capture observed correlations on multiple time scales, across sites, as well as relationships with temperatures (details can be found in ref. [13]).

The historical daily temperatures from the VAR model (described earlier) were matched with a synthetic sample of streamflow, HDD, and CDD produced using the Gaussian Copula approach described above. For any chosen synthetic year, HDD and CDD data for winter and spring months were matched with the historical record's HDD and CDD using mean squared error. The historical year with the closest match was selected as the basis for determining the daily flow fractions at each streamflow gauge site. This allowed years with higher winter and spring temperatures to experience earlier snowmelt, as has been observed in California[13].

Finally, the synthetic records of hydrometeorological variables (streamflow, solar irradiance, wind speed, and temperature) time series were translated into associated power system inputs. Using multivariate regression models fitted to historical data (and residuals then represented using VAR processes), the synthetic hydrometeorological data were used to create daily records of wind power generation via wind speeds, solar power generation via solar irradiance, and zonal electricity demand via temperatures and wind speeds. Hourly values were resampled from historical datasets maintained by Bonneville Power Administration and CAISO based on closest daily values and day of the year.

To calculate daily available hydropower generation, three different approaches were used. The synthetic streamflow records were passed through mass-balance hydrologic models of dams in the Columbia River basin and major storage reservoirs in California; also, a machine-learning representation of high altitude hydropower production in California was used; and a small portion of remaining hydropower capacity was also represented via scaled model outputs (details can be found in the Supplemental Information section labeled "Additional Information of CAPOW" and in ref. [13]). Daily amounts of available hydropower were then optimally dispatched on an hourly basis by the UC/ED models.

## Data availability
All data used to run the CAPOW model used in this study are online via Zenodo[31], as are all data used to create figures[43].

## Code availability
A stable version of the CAPOW model used in this study is online via Zenodo[31], as is all code used to create figures[43].

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

## Acknowledgements

This work was supported under the National Science Foundation's Coupled Natural-Human Systems (CNH2) program, award number 2009726.

## Author contributions

A.Z. Conceptualization, Methodology, Software, Validation, Formal analysis, Visualization, Writing original draft. J.D.K. Conceptualization, Methodology, Software, Supervision, Funding acquisition, Resources, Writing original draft. A.Y. Conceptualization, Methodology, Supervision, Funding acquisition, Writing original draft. P.W. Conceptualization, Methodology, Writing original draft. A.B. Formal Analysis, Writing original draft.

## Competing interests

The authors declare no competing interests.
