## [Peer Review File · Nature Communications]

U.S. West Coast droughts and heat waves exacerbate pollution inequality and can evade emission control policiesREVIEWER COMMENTS

Reviewer #1 (Remarks to the Author):

Thank you for the opportunity to review this manuscript. The paper examines the relationship between hydrometeorology and power system emissions, with a focus on (in)equities in impacted populations. The paper combines a 500-year hydrometeorology synthetic weather ensemble with a California and Pacific Northwest power system model. Key findings include extreme heat drives the worst 1-day air quality outcomes, and drought conditions drive the worst annual air quality outcomes. Local air pollutant taxes on power plants reduce damages on most days, but not in periods of high grid stress (i.e., during peak temperatures and therefore demand).

Drivers of power system emissions are summarized in the paper as already establishing a link between heat, drought, and increased emissions. As such, the novelty of the results is two-fold: (1) in county-level assessments of health damages and (2) in running their analysis over a 500-year synthetic weather ensemble. While county-level assessments of health damages are manifold, the link identified here to heat & drought is impactful, although unsurprising given that scarcity yields less flexibility in operations. This county-level impact assessment and the exposed inequities I also think are important for real-world decisionmakers, especially given recent extreme heat and drought and potentially worse conditions in the near future.

I have several questions and suggestions below. Overall, though, I think this manuscript is strong.

- There is generally a lack of validation in the model results and, more importantly, model inputs. Demand is a particularly key driver of the study results, but details on the demand prediction algorithm and its accuracy are lacking.
- It is unclear how CAPOW (the dispatch model) accounts for imports from neighboring regions. During periods of high grid stress, the focus of this paper, California imports would be a key resource that the state would rely on. CAPOW seems to optimize operations in the PNW, which can export power to CA, and also accounts for imports from the Southwest. How are these imports determined? Do they respond to stress within California? What about interconnections between CA and the rest of WECC, e.g. imported wind power from the Great Plains states?
- Several typos are in the paper (emissions, Fibure, etc.), and significant formatting errors in the PDF exist (figure cross-references).
- Line 135: is there uncertainty in daily grid performance, or variability? There is no forecasting happening here, so I would suggest here (and basically everywhere in the paper) variability is a more appropriate term.
- Line 143-144: please clarify 1000 is the max price.
- Figure 1: how do these values (the left three) compare to observed values?

Reviewer #2 (Remarks to the Author):

The present paper addresses the question of how some severe meteorological conditions (specifically droughts and heat waves) affect air emissions in the California power grid, and therefore air quality and human health in areas of California. The paper draws two major conclusions. The first, which I believe the authors view as the central conclusion of the paper, is that extreme weather events in California can lead to situations rendering otherwise effective pollution control policies ineffective. This has negative consequences for public health. My view is that this conclusion could be framed more effectively. The secondary conclusion, which follows from the first, is that the impacts of increased air pollution are borne disproportionately by minority populations in California. This conclusion is interesting but needs substantially more development.

The approach taken by the authors is to use historical weather and streamflow data to generate a large number of simulated realizations of hydrological conditions and electricity demand in California. These simulated conditions are then integrated with an operational model of the California power grid. Overall I find the modeling approach is creative and, in principle, sound. I do have some questions and suggestions:

*The authors could be more careful in how they describe the synthetic data that they generate. The authors describe generating a "500 year stochastic ensemble" which at first read I took to mean a 500-year time series. What I think the authors have actually done is to generate 500 synthetic meteorological years based on sampling 50 years of historical data. I infer from the text that these synthetic years are generated independently; whether any more granular time steps are also independent of one another (or whether there is some autocorrelation built into the simulation engine) was not totally clear to me.

*The output of this synthetic data generation is at the hourly time step, correct? The authors imply this but I don't think are specific. Do the authors have all historical data at this hourly time step or is there some estimation/imputation going on?

*I (and other readers) would appreciate some more information on CAPOW in the SI. These details are very important for interpreting the modeling output. For example, the paper is not clear on how the CAPOW model handles imports into California from British Columbia (which, unlike much of the Western hydro system, has substantial storage in the form of reservoirs). If CAPOW is limiting those transfers into California, this is naturally going to result in the model dispatching more highly-polluting fossil generation in California.

*The authors are clear in bounding their analysis to reflect (more or less) current conditions in the California grid. This is understandable but doesn't allow the authors to speak to how grid operators like California's, which are rapidly transitioning to renewable energy, should be planning their systems to minimize the kinds of impacts described in the paper. This is a clear area for future research...but honestly also limits the policy contribution of the present paper.

*The use of historical data to generate synthetic meteorological conditions is also understandable. It will also naturally raise the question as to whether future conditions will be stationary in a changing climate (in the sense of being described by the same distribution). This is important for the authors' primary conclusion. Examining Figure 6, it appears that in 99.5% of simulated days there is no policy conflict between pollution control and reliable electricity. That conclusion is based on sampling from historical distributions.

The primary conclusion of the authors nicely illustrates a conundrum that operators and other players in the Western US power grid (and probably other hydro-dependent grid operators as well) have understood for many years. When hydro availability is tight (either in absolute terms or relative to electric demand), grid operators are faced with a situation where they must either allow electricity shortages or utilize power plants that have some adverse social consequences in the form of high costs to consumers, high levels of air emissions, or both. California has faced this tradeoff before, perhaps most publicly during its energy crisis in 2000/2001. During that period, on multiple occasions air quality regulators permitted highly-polluting power plants to run in California in order to avoid blackouts, in violation of air quality regulations. There are likely other situations where emergency conditions on the power grid may lead to the overriding of short-term air pollution regulations.

I would not, however, characterize this as a failure of the air emissions policy. The "policy" that is being modeled is a price on specific air emissions, with a goal of reducing those air emissions. Many actual air quality policies (even those that involve pricing pollutants) also involve attainment targets that must be met. If the targets are met, the policy "succeeds" or has met its goal. I am not sure from the paper if these targets are incorporated into the CAPOW optimization problem. In the case of the modeled policy, it is not clear what the specific policy objective is, or what the authors mean by success versus failure.

More generally, what I believe is being observed in the modeling results is not the failure of a single policy but the resolution of a conflict between multiple policies - one related to air quality and one related to electric reliability (just as California faced during its power crisis). In the case of the modeled outputs the results in many cases are placing a higher weight on electric reliability than air quality because the marginal penalty for one MWh of unserved electricity is substantially larger than the damage penalty for serving that MWh of electricity. My suggestion to the authors is

to reframe this policy discussion in terms of tradeoffs rather than the absolute success or failure of one modeled policy in isolation.

To support the paper's second conclusion, that the impacts of the policy conflict between reliability and air quality fall disproportionately on minority populations, my suggestion to the authors is that they provide more spatial evidence of this conclusion. It's an important conclusion but in my view could be more substantiated. It would be nice to see, spatially, which parts of California are disproportionately impacted by degraded air quality in the name of electric reliability. Are these impacted counties the same ones that host the highly-polluting power plants being dispatched for reliability (those seem to be shown or suggested in Figure 2). Are these affected counties already suffering from poor air quality that is being made worse by these specific episodes? Figure 3 contains some analysis aimed at identifying the drivers of undesirable social impacts, but that appears to be aggregated over space as well as over time. (I think - I was not completely clear as to how this figure was generated.)

A final question that arises from the present paper is the extent to which there are spillover effects into surrounding states in the Western grid. These could arise if power plants in California degrade air quality in surrounding states, or if polluting power plants in other states are asked to provide emergency electricity to California.

Reviewer #3 (Remarks to the Author):

Summary:

The authors present a detailed study of spatially resolved (county-level) air-quality impacts from power plant operations in California using 500 years of synthetic weather data. The authors also evaluate several scenarios where generators are taxed on their emissions of local air pollutants, global air pollutants, or both to assess the effectiveness of policies targeting high-polluting power plants during various hydrometeorological conditions. I enjoyed reading this paper. Overall, I found it insightful and the work's value to be well highlighted. My only major comment is that the methodology surrounding the generation of the synthetic weather data could be more clearly explained. I look forward to seeing this work and further research by this team in publication.

General comments:

- My main comment centres around the explanation of methods to generate the 500-year synthetic weather ensemble. As a reader, I find this part of the paper challenging to follow and as a researcher, I'm unsure if I would be able to reproduce the methods presented here. I have a handful of comments here (in the line comments below) that I think would solve this issue.
- Ensure that all variables in equations are clearly identified in the text (e.g., what is I_n ?).
- In the manuscript submission downloaded from the reviewer portal, there seem to be some referencing errors for the Figures in the manuscript. Although not a consequential reviewer comment, it is something the authors should address before resubmitting.
- If not done already, all figures in the paper should be submitted in vector format to ensure they remain crisp when resizing/zooming.
- To make the legend of Figure 1 cleaner, the authors could consider simply having the line for the median and the gradient colour scale for the percentiles in a single block, noting e.g., the minimum and maximum (or whatever points they want to highlight).
- The y-axis of Figure 6 would be clearer with a title of "Change in Power Generation (GWh)" or similar. The caption for the figure should also indicate what the change in each generator group's average generation is relative to (L413-414), i.e., the untaxed baseline scenario.

Line comments:

- L89: There is an extra "year" in this sentence.
- L111: Remove the "at" in the parentheses to be consistent.
- L121-122: For clarity, switch "zonal market prices" and "hourly estimates of electricity generation and emissions at each generator".
- L126: Spelling error for Figure.
- L255: Should be "negatively correlated".

- L277: I think there is a missing "and" between "high electricity demand" and "limited hydropower".
- L359: Is this sentence referring to the low availability of hydropower, wind, and solar for both Zone c and Zone d days? Or is it that days in both zones share similar wind and solar profiles and have low hydro availability? The sentence isn't quite clear.
- L369: I think there is a missing "and" in this sentence.
- L597: Regarding the streamflow data from 1954-2008 - Were all data gathered to populate the 500-year synthetic ensemble from (or extrapolated to) 1970-2017? If so, what was the process for extrapolating the streamflow data for 2009-2017? If streamflow data was not taken for the same time period, what is the justification behind this?
- L604: Remove "daily" for clarity.
- L618-619: There is an extra "profile of the" in this sentence.
- L635: Missing the "d" before the comma.
- L641: Missing the "k" after the equal sign.
- L644: Missing the closing parenthesis.
- L670-672: This sentence is unclear.
- L673: The use of "an approach" is too vague. What is the approach? Is there citable support for this approach?
- L684-685: Can you provide a reference for the translation of the weather data into CAPOW inputs for the reader? (Mostly relevant for translating temperature and wind speed to electricity demand).
- L689: What years do these historical datasets cover? And do these datasets contain only electricity demand, or also wind and solar generation? This is unclear.
- L692: "Streamflows" should be singular.
- L692-695: I find these lines particularly unclear. The sentences should be separated, and the description of these methods should ensure the process is repeatable. The description here is, at best, vague. Are there citations for the modelling or methods employed here? If not, there needs to be a better description of the methods used (at a minimum, included in the SI). What "model" is being used to scale outputs for the "small portion" (how much?) of remaining hydropower?

Dear Reviewers,

Thank you to the reviewers for their careful consideration of our work and helpful comments regarding our manuscript, which we have found to be very helpful in improving the paper. Each comment/question has been addressed below. In addition, we have provided a version of the revised manuscript in which all changes are highlighted yellow.

REVIEWER COMMENTS

Reviewer #1 (Remarks to the Author):

Thank you for the opportunity to review this manuscript. The paper examines the relationship between hydrometeorology and power system emissions, with a focus on (in)equities in impacted populations. The paper combines a 500-year hydrometeorology synthetic weather ensemble with a California and Pacific Northwest power system model. Key findings include extreme heat drives the worst 1-day air quality outcomes, and drought conditions drive the worst annual air quality outcomes. Local air pollutant taxes on power plants reduce damages on most days, but not in periods of high grid stress (i.e., during peak temperatures and therefore demand).

Drivers of power system emissions are summarized in the paper as already establishing a link between heat, drought, and increased emissions. As such, the novelty of the results is two-fold: (1) in county-level assessments of health damages and (2) in running their analysis over a 500-year synthetic weather ensemble. While county-level assessments of health damages are manifold, the link identified here to heat & drought is impactful, although unsurprising given that scarcity yields less flexibility in operations. This county-level impact assessment and the exposed inequities I also think are important for real-world decisionmakers, especially given recent extreme heat and drought and potentially worse conditions in the near future.

I have several questions and suggestions below. Overall, though, I think this manuscript is strong.

- There is generally a lack of validation in the model results and, more importantly, model inputs. Demand is a particularly key driver of the study results, but details on the demand prediction algorithm and its accuracy are lacking.

We have added a significant amount of validation data related to statistical estimation of hourly demand in the CAPOW model. Please see Supplemental Information, section labeled 'Additional explanation of CAPOW'.

Note that this information (and a broader validation of the model) is available in Table 2 of the following paper:

Su, Y., Kern, J., Denaro, S., Hill, J., Reed, P., Sun, Y., Cohen, J., Characklis, G. (2020). "An open source model for quantifying risks in bulk electric power systems from spatially and

temporally correlated hydrometeorological processes” *Environmental Modelling and Software*. Vol. 126.

Synthetic time series of heating and cooling degree days and associated wind speeds for all 17 GHCN stations are used as independent variables in multivariate regressions of daily peak electricity demand. Separate models were used for each of the five WECC zones that comprise the core UC/ED problem in CAPOW. The multivariate regressions were trained on historical weather and electricity demand data over the period 2010-2016. This approach is able to produce accurate estimates of daily peak electricity demand (R^2 values range from 0.75 to 0.89 across the five zones), with demand in zones that experience lower heating and cooling needs being more difficult to represent using temperatures and wind speeds alone. After simulating daily peak electricity demand for each zone, hourly electricity demand is determined by multiplying peak demands with 24-hour load profiles for each zone and each calendar day. These profiles, which are calculated using historical data, represent the typical fraction of daily peak demand experienced in each hour.

- It is unclear how CAPOW (the dispatch model) accounts for imports from neighboring regions. During periods of high grid stress, the focus of this paper, California imports would be a key resource that the state would rely on. CAPOW seems to optimize operations in the PNW, which can export power to CA, and also accounts for imports from the Southwest. How are these imports determined? Do they respond to stress within California? What about interconnections between CA and the rest of WECC, e.g. imported wind power from the Great Plains states?

We have added a significant amount of validation data related to the treatment of CAISO imports in the CAPOW model. Please see Appendix, section labeled ‘Additional explanation of CAPOW’.

.

Note that this information (and a broader validation of the model) is available in Table 2 of the following paper:

- Su, Y., Kern, J., Denaro, S., Hill, J., Reed, P., Sun, Y., Cohen, J., Characklis, G. (2020). “An open source model for quantifying risks in bulk electric power systems from spatially and temporally correlated hydrometeorological processes” *Environmental Modelling and Software*. Vol. 126.

The same weather data used to model electricity demand (see response to previous comment), along with daily values of hydropower production and wind power production in the Pacific Northwest zone of the CAPOW model, are used as independent variables in multivariate regressions that estimate exchanges of electricity between the five WECC zones that make up the core UC/ED model and neighboring WECC systems that are not currently represented mechanistically. Imports/exports that are modeled statistically are identified in Figure S17 below (reproduced from Su et al., 2020, see reference above) as blue dotted lines and labeled in green.

Editorial Note: Figure adapted from Su, Y., Kern, J., Denaro, S., Hill, J., Reed, P., Sun, Y., Cohen, J., Characklis, G. (2020). "An open source model for quantifying risks in bulk electric power systems from spatially and temporally correlated hydrometeorological processes" *Environmental Modelling and Software*. Vol. 126. Copyright (2020), with permission from Elsevier

Figure S17, reproduced from Su et al. 2020. *EM&S*.

In most cases, these blue dotted lines represent specific aggregated transmission pathways that WECC refers to by number. Historically, power flows on some of these WECC paths have been bi-directional; however, most often electricity flows along these paths are "imported" by the core UC/ED model (red circles) from an adjacent zone outside the core UC/ED model (black circles). "Exports" describe the opposite; a demand for electricity in an outside zone, which must be satisfied by generators within the core UC/ED model. Import/export regressions, are trained on observed daily path flow data (typically assuming the classification of: imports = positive flow values; exports = negative flow values). Path flow data are available for the years 2010-2012. Modeled imports/exports show R^2 values ranging from 0.77 to 0.95, suggesting that daily regional exchanges of electricity can be represented statistically using weather and streamflow data, which collectively drive zonal electricity demand and the availability of wind and hydropower.

Note: Because we model imports/exports as a function of hydrometeorological conditions, our approach for modeling these processes does inherently "respond" to grid scarcity in California, at least to the degree that we observed over the training period (2010-2012). For example, periods of high demand in CAISO are positively correlated with imports into California from the Pacific Northwest (PNW) and Southwest; this correlation is also preserved in our creation of synthetic imports/exports.

However, exchanges between CAISO and other systems is not treated as a decision variable to be optimized. Instead, estimated exchanges are treated as a constraint on both the CAISO and separate PNW power systems model.

- Several typos are in the paper (emissions, Fibure, etc.), and significant formatting errors in the PDF exist (figure cross-references).

Thank you for pointing out this typos. We have corrected all of them.

- Line 135: is there uncertainty in daily grid performance, or variability? There is no forecasting happening here, so I would suggest here (and basically everywhere in the paper) variability is a more appropriate term.

We made several changes in the use of our terminology to distinguish between the causal effects of hydrometeorological variability on grid processes (e.g. variability in air temperatures → variability in demand) and uncertainty/risk regarding grid outcomes (e.g. what is the probability of the grid experiencing extreme scarcity, and/or what is the 95th percentile in daily air pollution damages experienced in July). In several cases, based on the reviewer’s comment, we changed our use of ‘uncertainty’ to ‘variability’. But we left the word ‘uncertainty’ in cases where that was more accurate.

- Line 143-144: please clarify 1000 is the max price.

Thank you for pointing this out, we have made the suggested change in the paper (see page 7).

- Figure 1: how do these values (the left three) compare to observed values?

We have added an additional figure (S27) and description to the Appendix, see section labeled ‘Additional explanation of CAPOW’. It is reproduced below for the purposes of responding to the reviewer.

Figure S27 from Supplemental Information.

These compare our probability plots of electricity demand, hydropower generation and market prices in CAISO to historical values. Historical data shown for electricity demand, market prices and hydropower are shown for 2019. Note that due to missing reported data from EIA from October 2019 to August 2020, the historical hydropower data shown drops to 0 after that point in the calendar year.

In general, historical CAISO electricity demand and hydropower production fall within the envelope of uncertainty represented by our 500-year probability plots.

Note as well that observed CAISO market prices go exceed the historical ensemble of prices in early 2019. In order to control for the specific impacts of hydrometeorological variability on system performance, our CAPOW runs deliberately keep the price of natural gas constant (and relatively low) across all 500 simulation years and throughout each year. In reality, natural gas prices along the U.S. West Coast increased in early 2019 due to abnormally cold weather in the Pacific Northwest, which increased demand for natural gas (and thus the price).

Reviewer #2 (Remarks to the Author):

The present paper addresses the question of how some severe meteorological conditions (specifically droughts and heat waves) affect air emissions in the California power grid, and therefore air quality and human health in areas of California. The paper draws two major conclusions. The first, which I believe the authors view as the central conclusion of the paper, is that extreme weather events in California can lead to situations rendering otherwise effective pollution control policies ineffective. This has negative consequences for public health. My view is that this conclusion could be framed more effectively. The secondary conclusion, which follows from the first, is that the impacts of increased air pollution are borne disproportionately by minority populations in California. This conclusion is interesting but needs substantially more development.

The approach taken by the authors is to use historical weather and streamflow data to generate a large number of simulated realizations of hydrological conditions and electricity demand in California. These simulated conditions are then integrated with an operational model of the California power grid. Overall I find the modeling approach is creative and, in principle, sound. I do have some questions and suggestions:

*The authors could be more careful in how they describe the synthetic data that they generate. The authors describe generating a "500 year stochastic ensemble" which at first read I took to mean a 500-year time series. What I think the authors have actually done is to generate 500 synthetic meteorological years based on sampling 50 years of historical data. I infer from the text that these synthetic years are generated independently; whether any more granular time steps are also independent of one another (or whether there is some autocorrelation built into the simulation engine) was not totally clear to me.

*The output of this synthetic data generation is at the hourly time step, correct? The authors imply this but I don't think are specific. Do the authors have all historical data at this hourly time step or is there some estimation/imputation going on?

*I (and other readers) would appreciate some more information on CAPOW in the SI. These details are very important for interpreting the modeling output. For example, the paper is not clear on how the CAPOW model handles imports into California from British Columbia (which,

unlike much of the Western hydro system, has substantial storage in the form of reservoirs). If CAPOW is limiting those transfers into California, this is naturally going to result in the model dispatching more highly-polluting fossil generation in California.

Thank you for emphasizing the need for a more in-depth description of the stochastic engine that produces the synthetic data products used by CAPOW. We have added more information about this to Supplemental Information, see section labeled, 'Additional explanation of CAPOW.'

The stochastic engine that is used to create inputs to the CAPOW model produces synthetic hydrologic data (both streamflow and estimates of hydropower availability) on a daily time step. Daily availabilities of hydropower then serve as inputs to the UC/ED module of CAPOW, which schedules hydropower production hourly as part of the cost minimizing mathematical program. Synthetic meteorological data (temperatures, solar irradiance, wind speeds) are also generated on a daily basis. We use this information to estimate daily values of peak electricity demand, daily solar and wind power production. Synthetic time series of heating and cooling degree days and associated wind speeds for all 17 GHCN stations are used as independent variables in multivariate regressions of daily peak electricity demand. Separate models were used for each of the five WECC zones that comprise the core UC/ED problem in CAPOW. The multivariate regressions were trained on historical weather and electricity demand data over the period 2010-2016. This approach is able to produce accurate estimates of daily peak electricity demand (R2 values range from 0.75 to 0.89 across the five zones), with demand in zones that experience lower heating and cooling needs being more difficult to represent using temperatures and wind speeds alone. After simulating daily peak electricity demand for each zone, hourly electricity demand is determined by multiplying peak demands with 24-hour load profiles for each zone and each calendar day. These profiles, which are calculated using historical data, represent the typical fraction of daily peak demand experienced in each hour.

The same weather data used to model electricity demand, along with daily values of hydropower production and wind power production in the Pacific Northwest zone of the CAPOW model, are used as independent variables in multivariate regressions that estimate exchanges of electricity between the five WECC zones that make up the core UC/ED model and neighboring WECC systems that are not currently represented mechanistically. Daily values of peak electricity demand, daily solar and wind power production are then disaggregated to an hourly time step using profiles sampled from historical hourly data. Historical data used as the basis for our synthetic generator are daily hydrometeorological data and hourly records of wind and solar power production and electricity demand.

Note that each year is generated by resampling from statistical distributions fitted to roughly 50 historic years and is independent from the rest of 500-year ensemble (i.e. the other 499 years). Within year, we go to very careful lengths to capture all relevant statistical dependencies in hydrometeorological processes (air temperatures, wind speeds, solar irradiance, streamflow). These include: seasonality, diurnal effects, daily and hourly autocorrelation, statistical moments, and cross correlations across variables and space. We also note that while our synthetic data successfully captures these dependencies, it does so while also producing combinatorial extremes (i.e. extremely high and low net load scenarios) that exist outside the limited historical record.

Editorial Note: Figure adapted from Su, Y., Kern, J., Denaro, S., Hill, J., Reed, P., Sun, Y., Cohen, J., Characklis, G. (2020). "An open source model for quantifying risks in bulk electric power systems from spatially and temporally correlated hydrometeorological processes" *Environmental Modelling and Software*. Vol. 126. Copyright (2020), with permission from Elsevier

Figure S17. This figure is reproduced from Su et.al ¹

Imports/exports that are modeled statistically are identified in Figure S17 (reproduced from Su et al., 2020) as blue dotted lines (labeled green). In most cases, these blue dotted lines represent specific aggregated transmission pathways that WECC refers to by number. Historically, power flows on some of these WECC paths have been bi-directional; however, most often electricity flows along these paths are “imported” by the core UC/ED model (red circles) from an adjacent zone outside the core UC/ED model (black circles). “Exports” describe the opposite; a demand for electricity in an outside zone, which must be satisfied by generators within the core UC/ED model. Import/export regressions, are trained on observed daily path flow data (typically assuming the classification of: imports = positive flow values; exports = negative flow values). Path flow data are available for the years 2010-2012. Modeled imports/exports show R2 values ranging from 0.77 to 0.95, suggesting that daily regional exchanges of electricity can be represented statistically using weather and streamflow data, which collectively drive zonal electricity demand and the availability of wind and hydropower.

Note: Because we model imports/exports as a function of hydrometeorological conditions, our approach for modeling these processes does inherently “respond” to grid scarcity in California, at least to the degree that we observed over the training period (2010-2012). For example, periods of high demand in CAISO are positively correlated with imports into California from the Pacific Northwest (PNW) and Southwest; this correlation is also preserved in our creation of synthetic imports/exports. However, exchanges between CAISO and other systems is not treated as a decision variable to be optimized. Instead, estimated exchanges are treated as a constraint on both the CAISO and separate PNW power systems model.

For an in depth explanation of the methods employed by the stochastic engine and data products created, please refer to Su et al. 2020

*The authors are clear in bounding their analysis to reflect (more or less) current conditions in the California grid. This is understandable but doesn't allow the authors to speak to how grid operators like California's, which are rapidly transitioning to renewable energy, should be planning their systems to minimize the kinds of impacts described in the paper. This is a clear area for future research...but honestly also limits the policy contribution of the present paper.

We are actively planning a follow-up study that addresses the role of grid decarbonization on air quality concerns (and vice versa). While there have been several previous attempts to show the co-benefits of low carbon energy sources for human health (including exposure to air pollution), we think further work should examine the importance of including measures of inequality (including air pollution exposure) in system design. Within a given decarbonization timeline, there may be many feasible alternative grid configurations. We hypothesize that the lowest cost (e.g. leveled cost of electricity) system design may not be the most equal (defined using both environmental and economic measures), and there may be some system defines that successfully balance multiple objectives. We hope that this type of analysis will be benefit the planning activities of a broader set of ISOs and utilities.

*The use of historical data to generate synthetic meteorological conditions is also understandable. It will also naturally raise the question as to whether future conditions will be stationary in a changing climate (in the sense of being described by the same distribution). This is important for the authors' primary conclusion. Examining Figure 6, it appears that in 99.5% of simulated days there is no policy conflict between pollution control and reliable electricity. That conclusion is based on sampling from historical distributions.

The reviewer is correct; our analysis assumes climate stationarity, relative to the historical dataset used as the basis for our stochastic engine (a period of roughly 50 years spanning 1958-2008).

As new data has become available in recent years, future studies focused on estimating near term risks will include updated hydrometeorological data. In the past, we have made use of multi-model ensembles of climate futures to evaluate the performance of the West Coast power grid:

- Hill, J., Kern, J.D, Rupp, D., Voisin, N., Characklis, G. (2021). "[The Effects of Climate Change on Interregional Electricity Market Dynamics on the U.S. West Coast](https://doi.org/10.1029/2021EF002400)" *Earth's Future*. Volume 9, Issue 12. <https://doi.org/10.1029/2021EF002400>

We plan to incorporate these climate futures in future work that also addresses projected long term changes in the make-up of the power grid.

To address the reviewers comment, we have added additional discussion of the limitations of our assumption of climate stationarity in the discussion.

The primary conclusion of the authors nicely illustrates a conundrum that operators and other players in the Western US power grid (and probably other hydro-dependent grid operators as

well) have understood for many years. When hydro availability is tight (either in absolute terms or relative to electric demand), grid operators are faced with a situation where they must either allow electricity shortages or utilize power plants that have some adverse social consequences in the form of high costs to consumers, high levels of air emissions, or both. California has faced this tradeoff before, perhaps most publicly during its energy crisis in 2000/2001. During that period, on multiple occasions air quality regulators permitted highly-polluting power plants to run in California in order to avoid blackouts, in violation of air quality regulations. There are likely other situations where emergency conditions on the power grid may lead to the overriding of short-term air pollution regulations.

I would not, however, characterize this as a failure of the air emissions policy. The "policy" that is being modeled is a price on specific air emissions, with a goal of reducing those air emissions. Many actual air quality policies (even those that involve pricing pollutants) also involve attainment targets that must be met. If the targets are met, the policy "succeeds" or has met its goal. I am not sure from the paper if these targets are incorporated into the CAPOW optimization problem. In the case of the modeled policy, it is not clear what the specific policy objective is, or what the authors mean by success versus failure.

More generally, what I believe is being observed in the modeling results is not the failure of a single policy but the resolution of a conflict between multiple policies - one related to air quality and one related to electric reliability (just as California faced during its power crisis). In the case of the modeled outputs the results in many cases are placing a higher weight on electric reliability than air quality because the marginal penalty for one MWh of unserved electricity is substantially larger than the damage penalty for serving that MWh of electricity. My suggestion to the authors is to reframe this policy discussion in terms of tradeoffs rather than the absolute success or failure of one modeled policy in isolation.

Thank you for this thorough and thoughtful comment. We agree with the reviewer that the term "failure" does not accurately describe the instances we observe in our solution when the pollution tax does not result in reduced air pollution damages. The reviewer is correct that we are not instituting a pollution limit or target emissions reduction, but rather incorporating negative externalities into the marginal cost of electricity production. By doing so, it almost always alters the minimum cost UC/ED solution (heavier polluting power plants are used less in order to minimize costs).

There are a few instances in which we observe where the minimum cost UC/ED solutions are the same both with and without the pollution tax. In our manuscript, we refer to these instances as "failures" of the emissions control policy (the tax). In reality, what is happening is that the mathematical program is rationally avoiding loss of load (in our model valued at \$1000/MWh) instead of avoiding air pollution damages (\ll \$1000/MWh). This, as you put, is not a failure.

We have altered our terminology and included this important perspective in throughout the paper, to provide necessary context for our analysis (see pages 4, 5, 19, 20).

To support the paper's second conclusion, that the impacts of the policy conflict between reliability and air quality fall disproportionately on minority populations, my suggestion to the authors is that they provide more spatial evidence of this conclusion. It's an important conclusion but in my view could be more substantiated. It would be nice to see, spatially, which parts of California are disproportionately impacted by degraded air quality in the name of electric reliability. Are these impacted counties the same ones that host the highly-polluting power plants being dispatched for reliability (those seem to be shown or suggested in Figure 2). Are

these affected counties already suffering from poor air quality that is being made worse by these specific episodes? Figure 3 contains some analysis aimed at identifying the drivers of undesirable social impacts, but that appears to be aggregated over space as well as over time. (I think - I was not completely clear as to how this figure was generated.)

We have developed a new figure that we think demonstrates this part of results (See page 12). It is shown below and can also be found in the Supplemental Information section (Figure S4).

All four panels (a-d) show information about power plants in the CAPOW model in 2D matrix form. Each row contains information about a specific county. Counties are sorted from top to bottom according to the percentage of each county's population that identifies as non-white. . Each column contains information about a specific power plant. Power plants are sorted from left to right according to each power plant's marginal cost (in \$/MWh).

In panel a), the colored pixels represent (x,y) ordered pairs of (county, power plant). A colored pixel means that a power plant (y) does exist in county x. The color signifies the air pollution damage rate caused by that power plant across all counties in CA (this is the same quantity as the local air tax identified for each power plant), with blue representing low damages per MWh produced, and red representing high. The vast majority of power plants in California is located in the 26 (out of a total of 58) majority (i.e. >50%) non-white counties.

In panel b) the colors of each pixel instead signify the median air pollution damage (in \$) delivered from power plant (y) to county (x) across the 500-year stochastic simulation. In general, each column (whether largely blue or red) shows a noticeable, red-ward shift (indicating a significant increase in air pollution damages) at the precise point on the y-axis where county demographics shift from majority white to majority non-white.

In panel c) the colors of each pixel instead signify air pollution damage (in \$) delivered from power plant (y) to county (x) in the year with the highest annual damages across the 500-year stochastic simulation. Again, we see a noticeable, red-ward shift (indicating a significant increase in air pollution damages) at the precise point on the y-axis where county demographics shift from majority white to majority non-white.

Taken together, this information strongly supports the finding that non-white counties are disproportionately impacted by power plant air pollution in both "normal" and extreme years.

Panel d) shows the difference between panel b) and panel c). In other words, it shows the impacts of large negative anomalies in streamflow (drought) and large positive anomalies in air temperatures associated experienced during the year with the highest annual damages. Again, we see the red-ward shift in the bottom half of the panel. This indicates that the increased damages associated with grid scarcity also disproportionately impact majority non-white counties.

A final question that arises from the present paper is the extent to which there are spillover effects into surrounding states in the Western grid. These could arise if power plants in California degrade air quality in surrounding states, or if polluting power plants in other states are asked to provide emergency electricity to California.

Thank you for this comment. Cascading interregional impacts (such as those mentioned by the reviewer) almost certainly exist. Within the West Coast system, our results strongly suggest that hydrologic drought in the Pacific Northwest (reductions in imported hydropower) causes higher air pollution damages in California (see Figure 3). Likewise, higher demand in California is likely to result in increased imports from other regions (Southwest), which could come from polluting

power plants. Increased emissions from power plants in California (e.g. during dry years) does increase damages to other regions. Some of these effects are shown in Figure 1 of our manuscript. But the vast majority of damages from power plants located in California are incurred by people living in California, so we limited our scope to that state alone.

Reviewer #3 (Remarks to the Author):

Summary:

The authors present a detailed study of spatially resolved (county-level) air-quality impacts from power plant operations in California using 500 years of synthetic weather data. The authors also evaluate several scenarios where generators are taxed on their emissions of local air pollutants, global air pollutants, or both to assess the effectiveness of policies targeting high-polluting power plants during various hydrometeorological conditions. I enjoyed reading this paper. Overall, I found it insightful and the work's value to be well highlighted. My only major comment is that the methodology surrounding the generation of the synthetic weather data could be more clearly explained. I look forward to seeing this work and further research by this team in publication.

General comments:

- My main comment centres around the explanation of methods to generate the 500-year synthetic weather ensemble. As a reader, I find this part of the paper challenging to follow and as a researcher, I'm unsure if I would be able to reproduce the methods presented here. I have a handful of comments here (in the line comments below) that I think would solve this issue.

We have added additional explanation of how the 500-year synthetic weather ensemble (and corresponding time series of electricity demand, hydropower production, and wind and solar power production) was generated to the Supplemental Information, section Additional explanation of CAPOW. In addition, we have addressed all your comments below.

- Ensure that all variables in equations are clearly identified in the text (e.g., what is I_n ?).

Thank you for pointing this out, we have made the suggested changes in the paper (pages 32, 33).

- In the manuscript submission downloaded from the reviewer portal, there seem to be some referencing errors for the Figures in the manuscript. Although not a consequential reviewer comment, it is something the authors should address before resubmitting.

We have corrected all referencing errors for the Figures in the manuscript.

- If not done already, all figures in the paper should be submitted in vector format to ensure they remain crisp when resizing/zooming.

We will make sure to submit figures in vector format for the revised submission.

- To make the legend of Figure 1 cleaner, the authors could consider simply having the line for the median and the gradient colour scale for the percentiles in a single block, noting e.g., the minimum and maximum (or whatever points they want to highlight).

We have adjusted the figure as follows:

- The y-axis of Figure 6 would be clearer with a title of “Change in Power Generation (GWh)” or similar. The caption for the figure should also indicate what the change in each generator group’s average generation is relative to (L413-414), i.e., the untaxed baseline scenario.

Thank you for pointing this out, we have made the suggested change in the paper.

Line comments:

- L89: There is an extra “year” in this sentence.
- L111: Remove the “at” in the parentheses to be consistent.
- L121-122: For clarity, switch “zonal market prices” and “hourly estimates of electricity generation and emissions at each generator”.
- L126: Spelling error for Figure.
- L255: Should be “negatively correlated”.
- L277: I think there is a missing “and” between “high electricity demand” and “limited hydropower”.
- L359: Is this sentence referring to the low availability of hydropower, wind, and solar for both Zone c and Zone d days? Or is it that days in both zones share similar wind and solar profiles and have low hydro availability? The sentence isn’t quite clear.

Thank you for pointing these out, we have made all the suggested changes in the paper.

- L369: I think there is a missing “and” in this sentence.
- L597: Regarding the streamflow data from 1954-2008 - Were all data gathered to populate the 500-year synthetic ensemble from (or extrapolated to) 1970-2017? If so, what was the process for extrapolating the streamflow data for 2009-2017? If streamflow data was not taken for the same time period, what is the justification behind this?
- L604: Remove “daily” for clarity.
- L618-619: There is an extra “profile of the” in this sentence.
- L635: Missing the “d” before the comma.
- L641: Missing the “k” after the equal sign.
- L644: Missing the closing parenthesis.
- L670-672: This sentence is unclear.
- L673: The use of “an approach” is too vague. What is the approach? Is there citable support for this approach?
- L684-685: Can you provide a reference for the translation of the weather data into CAPOW inputs for the reader? (Mostly relevant for translating temperature and wind speed to electricity demand).
- L689: What years do these historical datasets cover? And do these datasets contain only electricity demand, or also wind and solar generation? This is unclear.
- L692: “Streamflows” should be singular.

Thank you for pointing these out, we have made the suggested changes in the paper.

- L692-695: I find these lines particularly unclear. The sentences should be separated, and the description of these methods should ensure the process is repeatable. The description here is, at best, vague. Are there citations for the modelling or methods employed here? If not, there needs to be a better description of the methods used (at a minimum, included in the SI). What “model” is being used to scale outputs for the “small portion” (how much?) of remaining hydropower?

Thank you for this comment. We have added more detail in the Supplemental Information, see section labeled , “Additional explanation of CAPOW” regarding our simulation of daily hydropower production in CAPOW.

Note that this information (and validation of our approach) is available in the following paper:

- Su, Y., Kern, J., Denaro, S., Hill, J., Reed, P., Sun, Y., Cohen, J., Characklis, G. (2020). “An open source model for quantifying risks in bulk electric power systems from spatially and temporally correlated hydrometeorological processes” *Environmental Modelling and Software*. Vol. 126.

REVIEWERS' COMMENTS

Reviewer #1 (Remarks to the Author):

I thank the authors for their rigorous responses to my comments and questions. I have no further comments, and recommend publication.

Reviewer #2 (Remarks to the Author):

I have reviewed the revised manuscript and the responses to the three reviewers. Overall I think the authors have done an admirable job of responding to the reviewer comments. The methods section of the paper, in particular, is much more clear after reading the revised manuscript.

My remaining comments are for clarification only.

- Table 1 provides some high level evidence in support of the authors' conclusions about the distributive effects of hydro-meteo variability but this table is largely separate from the discussion of those distributive effects. Perhaps there is an argument for introducing Table 1 (or at least the relevant numbers) earlier in the text.

- Perhaps this sentence in the abstract could be edited for clarity: "Communities of color and communities with high pollution burden are disproportionately impacted by increased emissions from power plants during droughts and heat waves." My first suggestion would be to replace 'communities' with a more precise term like 'counties with majority nonwhite populations' so that it aligns better with the discussion in the text. The term 'community' may also mean different things to different readers so a more exact term may be helpful. My second suggestion or question is that the sentence seems to indicate that counties with majority nonwhite populations and those with the largest pollution burden are not necessarily the same. But I think the authors are arguing in the discussion around Figure S4 that these sets of counties have a lot of overlap.

- I see that the authors have avoided using the term "fail" to describe policy effectiveness in the title, but this term still appears in the discussion section. I would still describe this as a policy conflict rather than a failure or one policy evading another. After all, the observed conflict comes down to a choice of modeling parameters that reflect policy preferences. If the value of unserved energy in CAPOW were decreased to half a cent per kWh, then my guess is that the modeling results would look quite different and the reverse policy conflict would emerge; i.e. that the modeled decision-makers are favoring air quality more heavily than electric reliability.

Reviewer #3 (Remarks to the Author):

Thank you to the authors for providing detailed responses to all reviewer comments. I maintain my perspective from the previous review that this is a strong, well-written paper with insightful methods and results. The authors have thoughtfully addressed my and other reviewer comments concerning the clarity of the presented methodology, with the addition of a detailed section in the supplemental information.

I would like to see more depth in the discussion surrounding the implication of the results of this study. Namely, how might the results of this study influence/inform policymakers and the decisions they face surrounding minimising (disproportionate) adverse effects while maintaining energy services? The authors mention that the results could influence how grid operators, grid participants, and policymakers plan around both hydrometeorological extremes and the minimisation of air pollution damages but stop short of detailing what that influence might be or the change in practice that could result.

I have two further small comments. Firstly, a small note about typos (e.g., line 160 missing a

closing parenthesis, line 362 potencial, temperatures) and formatting (e.g., Figure 4 caption). Secondly, about the diverging colour scales on Figures 5, S4, and S11. Consider using a continuous (e.g., single hue) colour scale for these figures, as the diverging scale, I think falsely, gives the impression of a "good", "bad", and "neutral" values for each figure. Since the data are continuous in most cases (and only good or bad in relative terms), a continuous hue is more appropriate unless the authors indeed are highlighting a divergence. In which case, this should be emphasised more clearly in the results/discussion. The obvious exception is Figure S4 panel d), where increases, decreases, and no/little change are represented. (The authors could also consider representing this panel in relative terms, which might be more indicative than a dollar value of change).

REVIEWERS' COMMENTS

Reviewer #1 (Remarks to the Author):

I thank the authors for their rigorous responses to my comments and questions. I have no further comments, and recommend publication.

Thank you again for reviewing our work.

Reviewer #2 (Remarks to the Author):

I have reviewed the revised manuscript and the responses to the three reviewers. Overall I think the authors have done an admirable job of responding to the reviewer comments. The methods section of the paper, in particular, is much more clear after reading the revised manuscript.

My remaining comments are for clarification only.

- Table 1 provides some high level evidence in support of the authors' conclusions about the distributive effects of hydro-meteo variability but this table is largely separate from the discussion of those distributive effects. Perhaps there is an argument for introducing Table 1 (or at least the relevant numbers) earlier in the text.

Thank you for this comment. We have followed this suggestion, and now introduce Table 1 a little earlier in the manuscript when we discuss Figure 3 and the correlation between low hydro/high demand (dry/hot) years and inequality measures.

- Perhaps this sentence in the abstract could be edited for clarity: "Communities of color and communities with high pollution burden are disproportionately impacted by increased emissions from power plants during droughts and heat waves." My first suggestion would be to replace 'communities' with a more precise term like 'counties with majority nonwhite populations' so that it aligns better with the discussion in the text. The term 'community' may also mean different things to different readers so a more exact term may be helpful. My second suggestion or question is that the sentence seems to indicate that counties with majority nonwhite populations and those with the largest pollution burden are not necessarily the same. But I think the authors are arguing in the discussion around Figure S4 that these sets of counties have a lot of overlap.

Thank you for this comment. We have adjusted the language in the abstract accordingly.

- I see that the authors have avoided using the term "fail" to describe policy effectiveness in the title, but this term still appears in the discussion section. I would still describe this as a policy conflict rather than a failure or one policy evading another. After all, the observed conflict comes down to a choice of modeling parameters that reflect policy preferences. If the value of unserved energy in CAPOW were decreased to half a cent per kWh, then my guess is that the

modeling results would look quite different and the reverse policy conflict would emerge; i.e. that the modeled decision-makers are favoring air quality more heavily than electric reliability.

We meant to remove all instances of our use of the word 'fail' to describe the policy conflict. They've all been removed now, and we instead describe these instances as times when penalties do not adequately incentivize emissions reductions, given the difference in the value of unserved energy and air pollution damages per MWh in the model's objective function.

Reviewer #3 (Remarks to the Author):

Thank you to the authors for providing detailed responses to all reviewer comments. I maintain my perspective from the previous review that this is a strong, well-written paper with insightful methods and results. The authors have thoughtfully addressed my and other reviewer comments concerning the clarity of the presented methodology, with the addition of a detailed section in the supplemental information.

I would like to see more depth in the discussion surrounding the implication of the results of this study. Namely, how might the results of this study influence/inform policymakers and the decisions they face surrounding minimising (disproportionate) adverse effects while maintaining energy services? The authors mention that the results could influence how grid operators, grid participants, and policymakers plan around both hydrometeorological extremes and the minimisation of air pollution damages but stop short of detailing what that influence might be or the change in practice that could result.

Thank you for this comment. We have added additional discussion of specific interventions that can/should be informed by the results of our research.

I have two further small comments. Firstly, a small note about typos (e.g., line 160 missing a closing parenthesis, line 362 potential, temperatures) and formatting (e.g., Figure 4 caption).

Thank you for catching these errors, we've corrected them.

Secondly, about the diverging colour scales on Figures 5, S4, and S11. Consider using a continuous (e.g., single hue) colour scale for these figures, as the diverging scale, I think falsely, gives the impression of a "good", "bad", and "neutral" values for each figure. Since the data are continuous in most cases (and only good or bad in relative terms), a continuous hue is more appropriate unless the authors indeed are highlighting a divergence. In which case, this should be emphasised more clearly in the results/discussion. The obvious exception is Figure S4 panel d), where increases, decreases, and no/little change are represented. (The authors could also consider representing this panel in relative terms, which might be more indicative than a dollar value of change).

Thank you for this comment; we agree, and have changed these three figures accordingly.